# Identifiable Latent Bandits:
# Leveraging observational data for personalized decision-making

**Ahmet Zahid Balcıoğlu**                                                    *ahmet.balcioglu@chalmers.se*
*Department of Computer Science*
*Chalmers University of Technology*
*University of Gothenburg*

**Newton Mwai**
*Department of Computer Science*
*Chalmers University of Technology*
*University of Gothenburg*

**Emil Carlsson**
*Sleep Cycle*

**Fredrik D. Johansson**
*Department of Computer Science*
*Chalmers University of Technology*
*University of Gothenburg*

**Reviewed on OpenReview:** *https://openreview.net/forum?id=SvkZ76wKpu*

## Abstract

Sequential decision-making algorithms such as multi-armed bandits can find optimal personalized decisions, but are notoriously sample-hungry. In personalized medicine, for example, training a bandit from scratch for every patient is typically infeasible, as the number of trials required is much larger than the number of decision points for a single patient. To combat this, latent bandits offer rapid exploration and personalization beyond what context variables alone can offer, provided that a latent variable model of problem instances can be learned consistently. However, existing works give no guidance as to how such a model can be found. In this work, we propose an identifiable latent bandit framework that leads to optimal decision-making with a shorter exploration time than classical bandits by learning from historical records of decisions and outcomes. Our method is based on nonlinear independent component analysis that provably identifies representations from observational data sufficient to infer optimal actions in new bandit instances. We verify this strategy in simulated and semi-synthetic environments, showing substantial improvement over online and offline learning baselines when identifying conditions are satisfied.

## 1 Introduction

The goal of personalized decision-making is to find the actions best suited for specific individuals. For example, chronic diseases such as rheumatoid arthritis have dozens of therapy options after diagnosis (Singh et al., 2016) whose efficacy for a new patient is unknown, and need to be tried out sequentially, until an optimal match is found (Murphy et al., 2007).

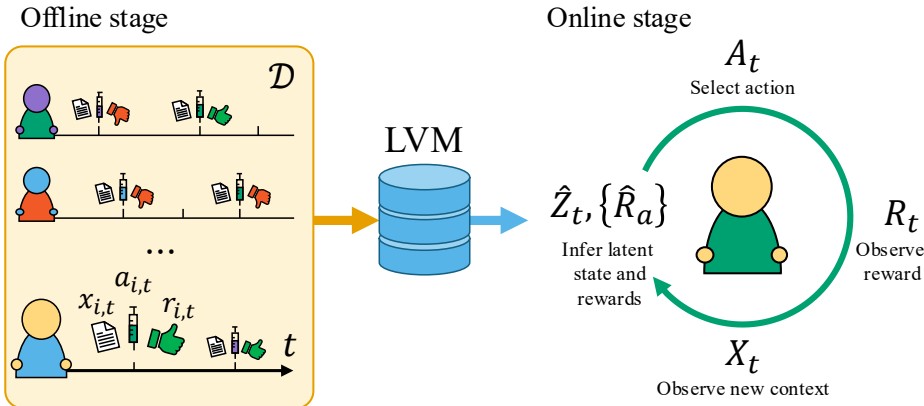

Figure 1: Identifying the best treatment for a new patient using ILB. Offline, we learn a provably identifiable latent variable model (LVM) (see Theorems 3.3 and 3.4), assumed known a priori in previous latent bandit algorithms. Online, we apply a decision-making algorithm making use of the LVM (see Algorithm 1).

Multi-armed bandits (MAB) Thompson (1933); Robbins (1952); Gittins (1979) have been studied extensively for online sequential decision-making of this form, but tend to require many more trials to converge than any single patient could go through, precluding their use in personalized medicine (Kinyanjui et al., 2023). Developing methods that exploit similarities between problem instances and reduce the necessary exploration is paramount.

A pragmatic solution to minimize the sample complexity in personalized decision-making is to leverage (offline) observational data of previous decisions and outcomes Rosenbaum et al. (2010). For example, estimating conditional causal effects Rubin (2005); Shalit et al. (2017); Künzel et al. (2019); Hahn et al. (2019) from cross-sectional or longitudinal observational data allows decision-makers to tailor choices to a set of context variables. However, a single set of context variables observed passively before making decisions is usually insufficient to identify an optimal *personalized* action, which may depend on unobservable factors.

Paradigms blending online and offline learning have been proposed to shorten exploration. For instance, contextual bandit algorithms exploit the structure between a *context variable*, actions, and rewards to personalize decisions (Chu et al., 2011; Agrawal & Goyal, 2013; Zhou, 2015; Lattimore & Szepesvári, 2020). Their sample complexity can be further reduced by either warm-starting model parameters for online learning by learning from historical data in an offline phase Zhang et al. (2019); Oetomo et al. (2023; 2024), or leveraging historical data to uncover structure through observable or latent clustering Bui et al. (2012); Bouneffouf et al. (2019); Maillard & Mannor (2014); Hong et al. (2020); Kinyanjui et al. (2023); Huch et al. (2024), matrix decomposition Sen et al. (2017), or spectral methods Kocák et al. (2020); Russo et al. (2024).

When problem instances obey a shared *latent structure*, known a priori, *latent bandits* (Hong et al., 2020; Kinyanjui et al., 2023) have proved theoretically and empirically more sample efficient than unstructured bandits, but leave a fundamental question open: *How can we learn such latent structure from data and when will it lead to optimal decision making?*

Provable recovery of such latent structure is the goal of identifiable representation learning, and is possible under structural assumptions, such as independent latent components Hyvarinen & Morioka (2016) or particular causal structure Schölkopf et al. (2021), using methods like normalizing flows Rezende & Mohamed (2015), or contrastive learning Gutmann & Hyvärinen (2010). We build on these developments to learn identifiable representations to improve personalized decision-making.

**Contributions.**  (1) We introduce *identifiable latent bandits*, ILB, the first family of latent bandit algorithms that recover a continuous vector-valued latent state without requiring the latent variable model (LVM) to be known a priori. (2) We build on nonlinear independent component analysis (ICA) Comon (1994) for identifiable representations and introduce mean-contrastive learning and use it to provably learn the LVM. (3) We prove that this framework is partially identifiable to a degree sufficient for optimal decision-making and

propose three algorithms that exploit the latent variable model for personalized sequential decision-making in the regret minimization setting. Our framework is summarized in Figure 1. (4) We show in experiments that, when the conditions of our theory hold, our algorithms are more sample-efficient than online bandits, less biased than offline (regression) baselines, and preferable to hybrid alternatives, both when a perfect (oracle) model is used and when the LVM has been learned from observational data. We test the sensitivity of our algorithm to various violated assumptions and demonstrate its efficacy in a semi-synthetic environment for choosing a therapy for patients with Alzheimer's disease Kinyanjui & Johansson (2022).

## 2 Problem setup

We use the choice of medical treatment as a running, motivating example and model the decision-making process for problem instance (patient) $i \in \mathbb{N}$ who takes actions (treatments) $A_{i,t} \in \mathcal{A} = \{1, ..., K\}$ over rounds $t = 1, ..., T$, resulting in observed stochastic rewards (responses) $R_{i,t} \in \mathbb{R}$ with means $\mu_{i,a_t}$. At each round, a decision-maker (physician) observes a set of context variables $X_{i,t} \in \mathbb{R}^d$ and aims to select an action $A_{i,t}$ based on the history $H_{i,t} = (X_{i,s}, A_{i,s}, R_{i,s})_{s=1}^{t-1}$ and the current context $X_{i,t}$, to minimize the cumulative regret ($\text{Reg}_T$) (Lattimore & Szepesvári, 2020),

$$\text{Reg}_T = \mathbb{E}\left[\sum_{t=1}^{T}(\mu_i^* - R_{i,t})\right] \quad \text{where} \quad \mu_i^* = \mu_{i,a^*}, \tag{1}$$

with $a_i^* = \arg\max_{a \in [K]} \mu_{i,a}$ the optimal action for the current problem instance. Without further assumptions, achieving small regret typically requires prohibitively many trials to learn the reward distributions for a new instance (Håkansson et al., 2020). To mitigate this, we exploit *shared structure* between the rewards of different instances so that previous instances can inform the solutions of future ones.

We assume the rewards are structured according to a *latent variable* $Z_i \in \mathbb{R}^n$, constant for each instance $i$, which fully determines the reward distribution of each action. Consequently, any two instances $i, j$ with the same latent state $z_i = z_j = z$ share expected rewards of actions $\mu_{i,a} = \mu_{j,a} = \mu_a(z)$, and optimal arms. The same assumption is central to *latent bandits* Maillard & Mannor (2014); Sen et al. (2017); Kinyanjui et al. (2023); Hong et al. (2020). The key components of latent bandit algorithms are a latent variable model (LVM) approximating $p(Z_i \mid H_{i,t}, X_{i,t})$ and a reward model $\mu_a(z)$ for each value of $z$, used to select the next action using a *selection criterion* based on an inferred value of $Z$. For example, the `mTS` algorithm Hong et al. (2020) samples $\hat{z}_t \sim p(Z_i \mid H_{i,t}, X_{i,t})$ and selects the action $a_t = \arg\max_a \mu_a(\hat{z}_t)$. However, this and related works assume that both models are known a priori but give little guidance for how to learn or acquire them.

We posit that in real-world applications algorithms must *learn* the LVM from *observational historical data* $\mathcal{D} = \{(x_{1,t}, a_{1,t}, r_{1,t})_{t=1}^{T_1}, ..., (x_{I,t}, a_{I,t}, r_{I,t})_{t=1}^{T_I}\}$ of $I$ previous problem instances, each with a sequence length $T_i$, $i \in [I]$. This raises a fundamental problem: multiple LVMs may fit the observed data $\mathcal{D}$ equally well yet differ in their estimated latent posterior $\hat{p}(Z|H_t, X_t)$, potentially leading to latent bandits recommending suboptimal actions. Such ambiguity is resolved, for sufficiently large data sets, when the data-generating process is *identifiable*—uniquely recoverable from the distribution of observable data. We discuss identifiability further in Appendix B.

Identifiability of the posterior of $Z$ alone is not sufficient for decision-making, as interventional reward distributions may not be identifiable. For example, if different patients represented in $\mathcal{D}$ were given systematically different treatments depending on an unobserved variable then $\mu_a(z) \neq \mathbb{E}[R_t|A_t = a, Z = z]$ in general. We view the reward of an action $a$ as the causal effect of an intervention $\mathbf{do}(A_t = a)$ (Pearl, 2009) on the instance, and define $\mu_a(z) := \mathbb{E}[R \mid \mathbf{do}(A_t = a), Z = z]$, where the $\mathbf{do}$-notation of Pearl (Pearl, 2009) distinguishes intervening from conditioning on the action $A_t$. This distinction is critical when learning $\mu_a(z)$ from observational data as this faces threats of confounding and other biases (Pearl, 2009). Thus, to apply a latent bandit algorithm in online decision-making, we must first show that both

$$\begin{align} &\text{i)} \ p(Z \mid H_t, X_t) \quad \text{and} \\ &\text{ii)} \ \mu_a(z) = \mathbb{E}[R_t \mid \mathbf{do}(A_t = a), Z = z] \end{align} \tag{2}$$

can be identified and estimated from $\mathcal{D}$:

The central goal of this work is to (i) design an algorithm that learns an *identifiable* model of the latent variable $Z$ and the rewards of actions $\mu_a(z)$ from observational data $\mathcal{D}$ during an *offline* phase, and (ii) prove that it leads to personalized, *online* sequential decision-making algorithms with lower sample complexity than algorithms that ignore $\mathcal{D}$.

## 2.1 Additional related work

Contextual bandits (Chu et al., 2011; Agrawal & Goyal, 2013; Zhou, 2015; Lattimore & Szepesvári, 2020) exploit structure in the rewards of actions by parameterizing their distribution as a function $\hat{\mu}_a(x)$ of an observed context, $x$ and apply these parameters in new contexts. The problem is distinct from ours and has a different goal. In contextual bandits, each context $x$ is associated with a potentially different optimal action and reward distribution. In our setting, the optimal action is the same in each round $t = 1, ..., T$, and a single context $X_t$ at any one round $t$ is insufficient to fully determine the optimal action. Thus, *contextual bandits do not solve our problem*. We give a closer comparison of latent and contextual bandits in Appendix F.

A large branch of causal inference research aims to estimate conditional causal effects of actions (CATE) from observational (offline) data to support future decision-making (Radcliffe, 2007; Athey & Imbens, 2015; Yao et al., 2021), even when only proxy variables are available for unseen confounders Cui et al. (2024). Representation learning can be used to predict causal effects more accurately by embedding high-dimensional covariates and actions in a space that reveals causal relations Shalit et al. (2017); Schölkopf et al. (2021); Wang & Jordan (2021) and latent variable models (Louizos et al., 2017; Rakesh et al., 2018; Lu et al., 2022; Zhong et al., 2022) can be used to recover from confounding due to unobserved variables by exploiting assumptions on the data generating process. Previous work viewing bandits as online causal effect estimators (Lattimore et al., 2016; Lee & Bareinboim, 2018; Bareinboim et al., 2015; Louizos et al., 2017) have also mostly focused on remedying the effects of unobserved confounders. However, unobserved confounding is not a focus here, and these works are not concerned with sample-efficient online decision-making.

# 3 Identifiable latent bandits

In this section, we give a two-stage latent bandit algorithm that combines offline and online learning to perform optimal personalized decision-making in the online stage. We prove that, under the right conditions (3.1), both a latent variable model (3.2) and decision-making criteria (3.2.1), can be learned from observational data in the offline stage. We use these results to give provably efficient sequential decision-making algorithms (3.3) for new problem instances in the online stage. We illustrate this approach, dubbed *identifiable latent bandits* (`ILB`), in Figure 1.

## 3.1 Identifying assumptions on the data-generating process

Our modeling assumptions are general but well-motivated by the problem of finding the right symptomatic treatment for patients with chronic disease. For such conditions, drugs only affect symptoms and cannot cure the disease ($Z$ is constant in time), and both symptoms and responses vary with time in unpredictable ways ($X_t$, $R_t$ are noisy). Hence, a single trial of each action is insufficient to identify the optimal personalized treatment for a patient Håkansson et al. (2020). Finally, we assume the rewards of actions for an instance are stationary and determined up to exogenous noise by the state $Z$, $p(R_t \mid Z, \mathbf{do}(A_t = a)) = p(R_{t'} \mid Z, \mathbf{do}(A_{t'} = a))$. This is plausible for conditions that do not progress more rapidly than treatment exploration is performed. We formalize this below.

**Assumption 3.1** (Identifying assumptions). As illustrated in Figure 2, we assume that

(a) Each instance $i$ is generated by the following structural equations, for all $t \in [T_i]$,

$$
\begin{aligned}
Z_i &= U & Z_{i,t} &= Z_i + \eta_{i,t} \\
X_{i,t} &= g(Z_{i,t}) & R_{i,t} &= \theta_{A_{i,t}}(Z_i) + \epsilon_{A_{i,t}}
\end{aligned}
\tag{3}
$$

where $\eta_{i,t} \sim \mathcal{N}(\mathbf{0}, \sigma^2\mathbb{I})$ and thus, each source variable $Z_{i,t} \in \mathbb{R}^n$ is stationary in time with respect to the instance $i \in [I]$.

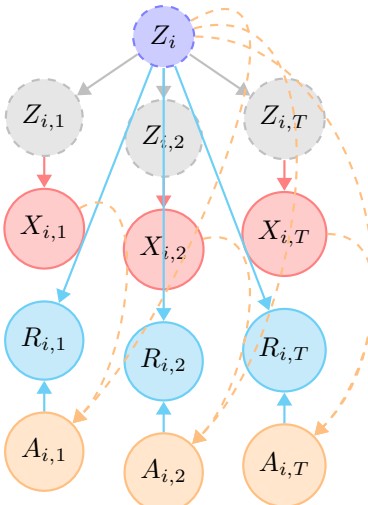

Figure 2: The structural causal model of Assumption 3.1 for an example patient instance $i$. Dashed arrows indicate potential sources of confounding bias that our model can handle.

(b) $U$ follows a non-parametric product distribution.

(c) The nonlinear transform $g$, referred to as the *emission function*, is smooth and injective.

(d) Rewards are generated according to a function $\theta_A$, noise $\epsilon_{A_{i,t}}$ is mean-zero Gaussian with variance $\sigma_A^2$.

We make no assumptions on the distribution of actions $A_{i,t}$ other than the causal (and probabilistic) independencies indicated in Figure 2. We assume that $\theta$ is a linear transformation for most of the discussion and simply denote it as a matrix. We investigate the nonlinear case empirically in Appendix E.5.

**How strong are the assumptions on $g$?** In prior works on latent bandits Hong et al. (2020); Zhou & Brunskill (2016); Maillard & Mannor (2014), the authors contend that LVM is estimated offline, but do not describe how such a model is learned. Some related works by Sen et al. (2017) and Kocák et al. (2020) assume a linear and spectral model for the LVM respectively. In comparison, our framework generalizes to nonlinear emission functions $g$. Assumption 3.1 c) is more typical of the literature on nonlinear ICA (Hyvarinen & Morioka, 2016) and is a necessary but not sufficient condition for the recovery of $Z$.

Under Assumption 3.1, the set of expected rewards is determined by the latent state through $\theta_A(z_i)$ per equation (3). Consequently, if $\theta_A$ is known, it is sufficient to infer $z_i$ to make an optimal decision for patient $i$. As we will see next, the assumption that context variables $X_1, \ldots, X_t$ are generated from a noisy $Z$ through an *injective* transform supports precisely this strategy.

### 3.2 Offline stage: Identifying and estimating the latent variable model

In the *offline stage* of the ILB framework, we learn the inverse of the emission function $g^{-1}$ and the reward model $\theta$ from the observational data $\mathcal{D}$ to support inferring the latent state $Z_i$ and the best possible action for a new instance $i$. As $g$ is injective we define $g^{-1}$ to be the left-inverse i.e $(g^{-1}(g(z)) = z$. To fit $g^{-1}$, we use contrastive learning with multinomial logistic regression inspired by identifiable representations literature (Hyvarinen & Morioka, 2016) where we learn from observed contexts $x_t \in \mathcal{D}$, stripped of instance identifiers, to predict to which instance $c \in [I]$ an observation belongs. Identifiable representation literature shares our goal of identifying the data generating distribution by inverting the emission function (Zimmermann et al., 2022). The standard assumption in that setting is that the observations differ in their noise and are assumed to arise from an unknown mixing of a known family of distributions (Hyvarinen & Morioka, 2016; Khemakhem et al., 2020b). In our setting each instance differs in mean instead, to emphasize this, we call the resulting contrastive learning algorithm *mean-contrastive*.

We fit a deep feature extractor $f : \mathbb{R}^d \to \mathbb{R}^n$ and a multinomial logistic regression model with softmax activation logits over classes $c$ given by $q_c(f(x)) = W_c^\top f(x) + b_c$, yielding the classifier

$$p(C = c \mid X = x; W, b) = \frac{e^{q_c(f(x))}}{1 + \sum_{j=2}^{I} e^{q_j(f(x))}}, \tag{4}$$

where $C$ is the instance indicator and $W_c \in \mathbb{R}^n$ and $b_c \in \mathbb{R}^n$ are instance-specific weights and biases respectively. We say that a feature extractor $f^*$ is optimal if there is a classifier based on $f^*$ that maximizes the resulting log-likelihood

$$f^*, q^* = \arg\max_{f,q} \sum_{i=1}^{I} p(C = i) \, \mathbb{E}_{X|C=i} \left[ \log p(C = i \mid X = x; f, q) \mid C = i \right]. \tag{5}$$

A universal function approximator on the form of (4) with feature extractor $f^*$ maximizes the log-likelihood (5) in the infinite-sample limit and converges to the true posterior $p(C \mid X)$, see Lemma B.3 in Appendix B. For the learning of such a feature extractor and classifier to be viable, we make the following assumptions:

**Assumption 3.2** (Viable learning task). We assume the following for the learning problem in (5).

(a) The dimension of the latent state is known and less than or equal to the feature extractor $f$ i.e. $n \leq d$.

(b) The matrix of patient latent states for instances in $\mathcal{D}$, $M = [z_1, ..., z_I]^\top \in \mathbb{R}^{n \times I}$ has rank $n$; that is, patients are sufficiently distinct.

Assumption 3.2 (a) is a simpler statement of our earlier assumption Assumption 3.1 (c) of injective data generating process. The second assumption relates to the variation of instances in the dataset, as we can not expect to fully recover the latents if the variation is not reflected in the dataset. We can now state our identifiability result.

**Theorem 3.3** (Identifiability of inverse emission function). *Under Assumptions 3.1–3.2 the optimal feature extractor $f^\star$, according to (5), is equal to the inverse emission function $g^{-1}$ up to an invertible affine transformation. In other words,*

$$Bf^\star(x) + b = g^{-1}(x) \tag{6}$$

*for constant invertible matrix $B \in \mathbb{R}^{n \times n}$, and $b \in \mathbb{R}^n$.*

We give the proof of Theorem 3.3 in Appendix B.1. The result partially identifies $p(Z \mid H_t, X_t)$, (i) in Equation (2), as the distribution of the true latent state $Z = z$ is Gaussian around the inverse emission function $g^{-1}$ by Assumption 3.1, i.e. $z = \mathbb{E}[g^{-1}(X_t)|Z = z]$. This allows us to have an unbiased estimate of $z$, up to an affine transformation, using $\mathbb{E}[f(X_t)|Z = z]$. Such affine identifiability results are common in the literature and make extensive use of the parametric form of the latent distribution Hyvarinen & Morioka (2016); Khemakhem et al. (2020a). Fitting $f$ by solving (5) forms the first step of Algorithm 1.

### 3.2.1 Identifiability of reward model and decision-making criteria

Once the feature extractor $f$ has been learned, we can estimate the patient latent state as $\bar{z}_i = \frac{1}{T_i} \sum_{t \in [T_i]} f(x_{i,t})$, and the reward model $\theta_a$ for each action $a$ using regression fit to input-output pairs $(\bar{z}_i, r_{i,t})$ using observations $(x_{i,t}, a_{i,t}, r_{i,t}) \in \mathcal{D}$ where $a_{i,t} = a$ (see Line 2 of Algorithm 1). Theorem 3.3 only guarantees that $f$ is an accurate model of $g^{-1}$ *up to an affine transform*. However, as we prove below this is sufficient to identify $\mathbb{E}[R_{i,t} \mid Z_i = z, \mathbf{do}(A_{i,t} = a)]$.

**Theorem 3.4.** *Assume that reward means are linear, $\mu_a(z) = \theta_a^\top z$, and fix a problem instance $i$. Then, under the conditions of Assumption 3.1, the state-conditional expected reward $\mathbb{E}[R_{i,t} \mid Z_i = z, \mathbf{do}(A_{i,t} = a)]$ of an intervention $a$ is identifiable from the observational distribution of problem instances $p(H_T)$ by the OLS regression estimand applied to observed rewards and latent states inferred by an optimal feature extractor $f$ in the sense of Theorem 3.3.*

---

**Algorithm 1** Identifiable latent bandits (`ILB`) with `CPG` and `FPG` decision-making criteria

---

**Observational data**: Learn LVM

1: Use observational data $\{(x_{i,t}, c_{i,t})\}_{i\in[I],t\in[T_i]}$ with $c_{i,t} := i$ the instance index to train the contrastive learning model $f$.
2: Fit $\hat{\theta}$ to inferred latent states $\bar{z}_i$ and rewards in $\mathcal{D}$ using OLS.

**Decision-making time:** Infer $Z$

1: **for** $t = 1, \ldots, T$ **do**
2:      Observe new context $x_{i,t}$ for instance $i$
3:      Use LVM estimate $\hat{z}_{i,t} = \frac{1}{t}\sum_{t'=1}^{t} f(x_{i,t'})$
4:      **if** `CPG`: Update belief about latent state
         $\hat{z}_i := \hat{\mathbb{E}}[z_i | x_{i,1}, \ldots, x_{i,t}] = \hat{z}_{i,t}$
5:      **if** `FPG`: Update belief about latent state
         $\hat{z}_i := \hat{\mathbb{E}}[z_i | h_{i,t}, x_{i,t}] = \arg\min_z \left[ \|z - \hat{z}_{i,t}\|^2 + \sum_{t'=1}^{t}(r_{i,t'} - \hat{\theta}_{a_{i,t'}}^\top z)^2 \right]$   (7)
6:      **if Greedy**:
7:        Estimate reward means, $\hat{\mu}_a = \hat{\theta}_a^\top \hat{z}_i$.
8:      **if Exploration:** (for `FPG` only)
9:        Sample reward means, $\hat{\mu}_a \sim \mathcal{N}(\theta_a^\top \hat{z}_i, \hat{\theta}_a^\top V^{-1}\theta_a)$, for $V = \sum_{t'=1}^{t} \hat{\theta}_{a_{t'}} \hat{\theta}_{a_{t'}}^\top$.
10:     Choose next action as $a_{i,t} = \arg\max_{a\in\mathcal{A}} \hat{\mu}_a$ and observe reward $r_{i,t}$
11: **end for**

---

The proof of Theorem 3.4 is given in Appendix C. The result shows the causal identifiability of the reward distribution, part ii) of the desiderata we laid out in Equation (2). It is applicable for cases where $Z$ is a valid adjustment set, such as when $X_t$ affects the treatment $A_t$. We show different confounding biases our approach can handle in Figure 2. We use Theorem 3.4 to derive the action selection criteria on lines 7 and 9 in Algorithm 1.

**Remark 1** (Remark on Theorem 3.4)**.** *The affine transformations of $\hat{Z} = \frac{1}{t}\sum_{t'} f(X_{t'})$ do not affect the ordering of $\hat{\theta}_A^\top \hat{Z}$ as long as reward parameters $\hat{\theta}_a$ are fit to this estimate. More explicitly, if we consider two actions $a_1$ and $a_2$, then for any fixed $z = B\hat{z} + b$,*

$$\theta_{a_1}^\top z > \theta_{a_2}^\top z \iff \tilde{\theta}_{a_1}^\top \hat{z} + \tilde{b}_{a_1} > \tilde{\theta}_{a_2}^\top \hat{z} + \tilde{b}_{a_2},$$

*where $\tilde{\theta}_a = \theta_a^\top B$ and $\tilde{b}_a = \theta_a^\top b$. This is because $B$ is invertible, so it induces a one-to-one transformation between $Z$, and $\hat{z}$ and does not affect the relative ordering of actions. Neither does the additive term $\tilde{b}_a$ affect the relative ordering of actions. Consequently,*

$$a^*(z) = \arg\max_a \theta_a^\top z = \arg\max_a \left[ (\theta_a^\top B)\hat{z} + \theta_a^\top b \right].$$

*Therefore, the optimal policy satisfies:*

$$a^*(z) = a^*(B\hat{z} + b) = a^*(\hat{z}).$$

### 3.3 Online stage: Estimation of the latent state & decision making

The results in Theorems 3.3–3.4 allow for an unbiased estimation of the rewards for each arm through an estimation of the latent state $\hat{\mu}_{i,a} = \hat{\theta}_a^\top \bar{z}_i$. In Algorithm 1, we present two different approaches for exploiting the learned LVM to *estimate* the (posterior distribution of the) latent state and show how our method allows for *exploration* in the decision-making.

In the online stage, a single context $x_{i,t}$ is noisy and does not carry enough information to accurately infer the instance-specific latent state $z_i$. However, under the conditions of Theorem 3.3, with an inference function $f$ that is optimal w.r.t. (5), $\hat{z}_{i,t} = \frac{1}{t}\sum_{t'=1}^{t} f(x_{i,t'})$ is an unbiased estimate of the latent variable $z_i$, up to a constant affine transform. Moreover, Theorem 3.4, justifies estimating $\mu_i$ by a linear model fit to $\hat{z}_{i,t}$. Thus, for a well-specified and well-estimated LVM, an intuitive approach is to play the best arm given the current

estimate $\hat{z}_{i,t}$ and previously estimated reward parameters $\hat{\theta}$ at each time-step, $a_t = \arg\max_{a \in \mathcal{A}} \hat{\theta}_a^\top \hat{z}_{i,t}$. We call this model *context posterior greedy* (CPG), as it uses only the context variables for posterior inference.

Despite its simplicity, the CPG algorithm has constant regret with respect to the horizon $T$ for LVMs with well-specified reward models. The regret bound also scales linearly in both the number of arms and the variance of the source variable, which is expected.

**Theorem 3.5.** *For an instance $i$, let $\Delta_i > 0$ such that $\forall a \neq a^* : \left| (\theta_{a^*} - \theta_a)^\top z_i \right| > \Delta_i$, and let $\overline{\Delta}_i = \max_{a \neq a^*} (\theta_{a^*} - \theta_a)^\top z_i$, and assume w.l.o.g. that $\forall a : \|\theta_a\|_2 = 1$. Then for a learned model pair $(f, \hat{\theta})$ such that $\forall a, z : \left| \hat{\theta}^\top f(g(z)) - \theta^\top z \right| < \epsilon$, the expected regret of CPG is bounded by*

$$
\mathrm{Reg}_T \leq \mathbb{1}_{[\epsilon < \frac{\Delta_i}{2}]} \frac{8K\sigma^2 \overline{\Delta}_i}{(\Delta_i - 2\epsilon)^2} + \mathbb{1}_{[\epsilon \geq \frac{\Delta_i}{2}]} \left( \frac{8K\sigma^2 \overline{\Delta}_i}{\Delta_i^2} + 2\epsilon T + \frac{\Delta_i T}{2} \right) \tag{8}
$$

*where $\sigma^2$ is the variance of $\eta_{i,t}$ (of $Z_{i,t}$ given $Z_i$).*

Theorem 3.5 is proven in Appendix D, we give a comparison to the setting of Hong et al. (2020) in Appendix D. The regret is constant for an optimal pair ($\epsilon = 0$), and linear when $\epsilon$ grows large and CPG becomes unable to distinguish between the arms in the $2\epsilon$ region. The bound *does not* depend on the magnitude of noise in the rewards as CPG is greedy with respect to the context. However, CPG fails to exploit the association between the latent state $z_i$ and rewards $r_{i,t}$, and can take longer to converge when the noise in the latent state is high or for out-of-distribution instances (see Appendix E.2).

In the case that the LVM is misspecified or misestimated, greedy reward maximization with respect to $\hat{z}_i$ as in Algorithm 1 will be biased in general and yield linear cumulative regret. In this case, one could either try to re-estimate the arm parameters $\theta_a$, or update the latent variable estimate $\hat{z}_i$ at inference time. These choices are equivalent as the reward is bilinear. An example of the latter is to look at the reward history and search for the latent $\hat{z}_i$ which best explains the previous rewards and contexts, conditioned on arm parameters. This trades off exploiting the inference function $f$ and explaining previous rewards. Our second algorithm, *full posterior greedy* (FPG), takes this approach and minimizes the full negative log-likelihood (7) to choose the best action. We recommend FPG as a more practical and adaptive alternative to CPG. In Lemma D.1, we analyze the mean and variance for FPG estimates of $z_i$.

**Exploration**   Once a latent is estimated, we could either use a greedy strategy and choose the best arm under the estimated latent state, or use the posterior for the estimated reward $\hat{\mu}_i = \hat{\theta}^\top \hat{z}_i$ for exploration. For FPG this can lead to choosing arms that reduce the variance of the estimate (7) and recover from a biased LVM estimate. We showcase an example of Thompson sampling on the posterior in line 9 of Algorithm 1, exploiting the fact that when the latent and reward noise is Gaussian, the estimated reward also follows a Gaussian distribution. We style this variant as FPG-TS. In Appendix E.1, we demonstrate the robustness of FPG and FPG-TS to LVM misspecification.

## 4   Experiments

**Environments**   We evaluate ILB in two settings, first a Synthetic environment, obeying the structural equations in (3), with a multivariate standard Normal for $U$, and $\theta_a$'s sampled from a centralized multivariate Normal distribution, normalized to unit vectors to ensure that the optimal treatment varies with $Z$. For the nonlinear mixing function $g$, we use a randomly initialized MLP with invertible square matrices and leaky ReLU activations to ensure invertibility. We sample treatments uniformly for observational data, $\mathcal{D}$. At inference, we average results over 100 instances with different latent states, generated from the same process. To test the sensitivity to problem parameters, we generate data with different sequence lengths $T_o$ and with different layers $L$ in the generating and fitting LVMs.

As a second environment, we use ADCB (Kinyanjui & Johansson, 2022), a simulator of Alzheimer's disease treatment. We modify the ADCB causal graph so that the latent state has categorical and continuous components, $Z_i = \{Z_i^\dagger, Z_i^{\mathtt{cat}}\}$ where the categorical components $Z_i^{\mathtt{cat}}$ comprise race and sex indicators, and

Table 1: LVM fitting. $L$ layers in the MLP, $T_o = 200$ time steps per instance. Mean correlation coefficient (MCC) for $\hat{z}$, average $R^2$ for reward estimates, and % correctly identified $a^*$.

| $L$ | Model | $\text{MCC}_Z$ | $R^2_R$ | % $a^*$ |
|---|---|---|---|---|
| **Synthetic environment** | | | | |
| 2 | LVM | 0.89 | 0.78 | 84 |
| 4 | LVM | 0.90 | 0.75 | 80 |
| 2 | VAE | 0.90 | 0.72 | 82 |
| 4 | VAE | 0.85 | 0.62 | 48 |
| 2 | Regression | - | 0.75 | 62 |
| 4 | Regression | - | 0.63 | 52 |

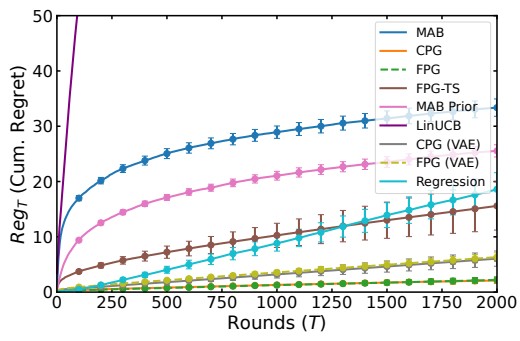

Figure 3: Cumulative regret results for ADCB, comparing `ILB` decision-making algorithms to baselines. Error bars indicate one standard error computed with 200 seeds. The LVMs are fitted across $I = 100$ instances with $T_o = 200$ points each with $L = 2$ layered model.

the continuous component $Z_i^\dagger$ comprises the ratio of Amyloid-$\beta$ (A$\beta$) plaques. The observed context ($X_{i,t}$) is a nine-dimensional mix of continuous and categorical values, generated from $Z_i^{\texttt{cat}}$ as well as a noisy continuous component $Z_{i,t}^\dagger$ with $\eta_{i,t} \sim \mathcal{N}(0, 0.02)$. Eight treatments are used, with a uniformly random observational policy, and conditional rewards are further described in Appendix G. We also noticed that the context *Age* variable has a unique value for each patient, which makes it trivial to predict the patient index and influences the `ILB` algorithm. We conduct additional experiments with the *Age* variable removed in Appendix E.3.

**LVM-based algorithms** For our identifiable LVM, we follow the network architecture and training procedure of Hyvarinen & Morioka (2016) and use an MLP for the feature extractor $f$ with hidden features equal in dimension with $Z_i$, as per Theorem 3.3. As an alternative to our identifiable LVM, we use the well-known variational autoencoder $\beta$-VAE Higgins et al. (2017), and adapt an implementation from PyOD repository Han et al. (2022). We specify training details for both models in Appendix H. After training either LVM, we use the observational data to estimate the reward parameter $\theta$. We apply decision-making algorithms `CPG`, `FPG`, and `FPG-TS` with both LVMs, as well as to the ground-truth inverse emission and reward models $g^{-1}, \theta$, referred to as "oracle". The oracle can be seen as providing the "true" model assumed known in previous latent bandit works (Hong et al., 2020).

**Bandit baselines** We compare to three online bandit algorithms. First, a Thompson sampling MAB (Thompson, 1933; Russo et al., 2018) that is oblivious to the latent state structure, initialized with Gaussian priors and with the ground truth variance of the reward. Second, an equivalent MAB, initialized with a one-shot prediction of rewards from the identifiable LVM, called MABPrior. Finally, we compare to a contextual upper confidence bound algorithm (LinUCB) Li et al. (2010).

**Regression baseline** We construct a Regression-based algorithm that ignores the latent structure and plays the action that maximizes an estimate of the expected reward $\mathbb{E}[R|A = a, X_1, ..., X_t]$ given the history of contexts, similar to `CPG`. The contexts are sufficient adjustment sets since actions are not confounded in $\mathcal{D}$ and the criterion is optimal given sufficiently long history. This baseline represents decision-making based on a causal effect estimator using a TARNet architecture (Shalit et al., 2017). See Appendix H for details.

**Evaluating LVM fit** We evaluate the fits of the LVMs on the Synthetic environment using the mean correlation coefficient (MCC) used in the ICA literature (Hyvarinen & Morioka, 2016), between the true latent states $Z_i$ and the recovered latents $\hat{z}_i$, on a held-out test set of 50 problem instances. To assess the reward model, we report the $R^2$ score between estimated and true potential rewards. We also look at the predicted best action and report the percentage of decision points where it equals the optimal action. The results are shown in Table 1. Both LVMs have high MCC scores across settings, suggesting that they are

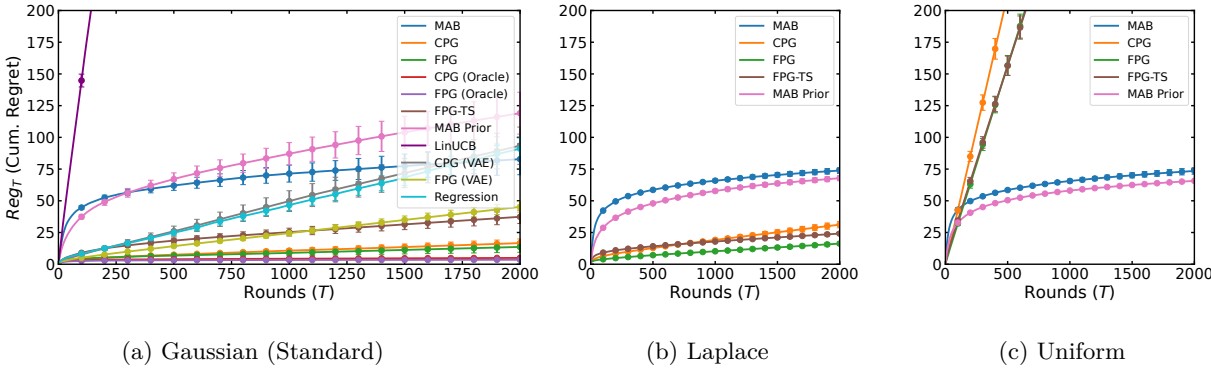

(a) Gaussian (Standard)        (b) Laplace        (c) Uniform

Figure 4: Cumulative regret for the Synthetic environment (left) comparing `ILB` decision-making algorithms to baselines, and comparative performance of our algorithm under different exponential noise (see Appendix E.6 for details). Error bars represent one standard error computed from 200 seeds. The LVMs are fitted across $I = 100$ instances with $T_i = 200$ time points each with $L = 2$ layered model.

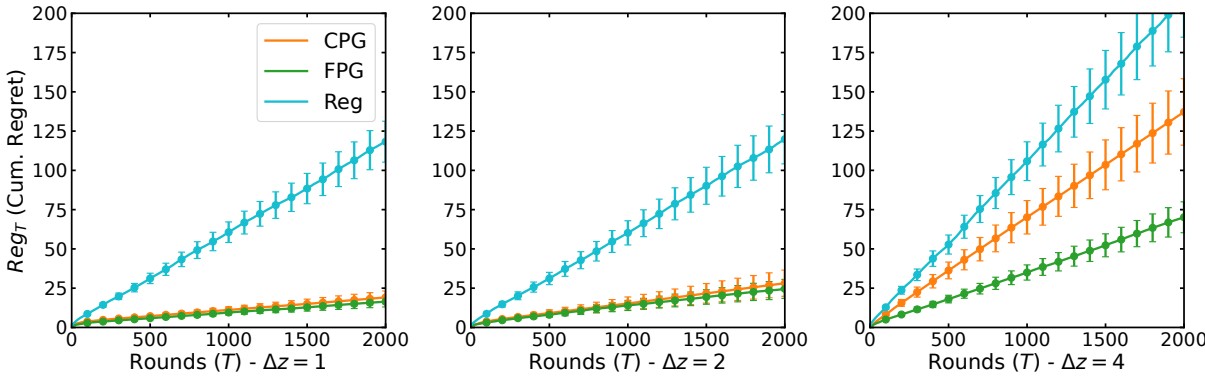

Figure 5: Cumulative regret for out-of-distribution experiments with increased $\Delta z$ difference from the training distribution on the synthetic data. Error bars indicate standard error over 200 seeds.

successful at inverting the encoding function $g$. Increasing the number of layers in the mixing MLP makes the learning and recovery tasks more difficult and affects the VAE more strongly. We select the $L = 2$ case for our bandit runs as all models have higher test-set $R^2$ here, to analyze the pros and cons of using each model in decision making. The Regression model has a comparable $R^2$ to the LVMs but has a lower rate of identifying the best action in the test set. All models perform comparably on ADCB (see Appendix I).

**Code and data availability**   We release the full implementation of `ILB` and code for reproducing the experiments at `https://github.com/Healthy-AI/identifiable-latent-bandits-public`. The repository additionally contains the generated ADCB data used in our experiments.

**Results for sequential decision-making**

**It is possible to learn effective latent bandits**   The results for all decision-making algorithms on the Synthetic environment are presented as regret plots in Figure 4, and the results for ADCB in Figure 3. In both cases, offline (Regression) and hybrid (`CPG`, `FPG`, `FPG-TS`) algorithms converge substantially quicker than the fully-online learners (MAB, MAB Prior, LinUCB), as expected. In Figure 4 (left), we see that the Oracle methods—latent bandits with a known, perfect model—perform the best, and that `CPG` and `FPG` (using the fitted `ILB` LVM) incur a very small bias. This confirms that *latent bandits are feasible to learn from historical data, without oracle access to the LVM.* Moreover, in the Synthetic environment, `CPG`

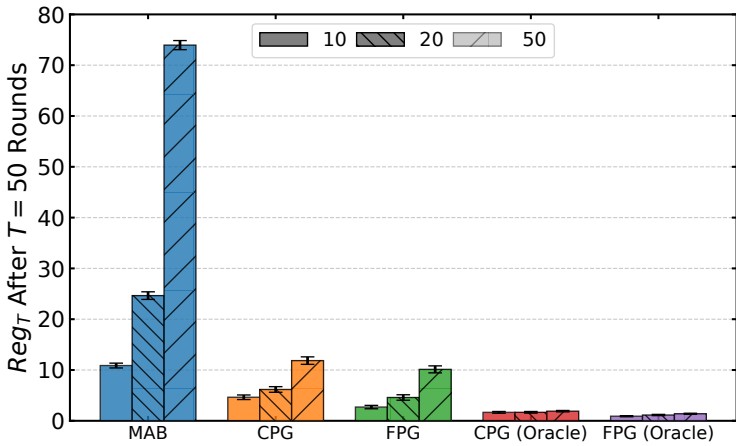

Figure 6: Expected cumulative regret for bandit algorithms in the cases of $K = 10$, $K = 20$, and $K = 50$ arms. Error bars show 1 standard error computed across 1000 seeds.

and `FPG` compare favorably to their equivalents using the VAE LVM. This is expected since `CPG`, `FPG` use well-specified reward models, matching the parameterization of environment. In ADCB, we also see evidence that a well-learned non-identifiable LVM (the VAE LVM has higher test $R^2$ in Table 5) can still yield quick convergence compared to online-only methods, seen in `CPG` (VAE) in Figure 3.

**Hybrid algorithms can overcome small bias in the LVM** `FPG-TS` is based on the same LVM as `CPG` and `FPG` and samples the reward model based on the full posterior of the latent state. It converges more slowly than `CPG` and `FPG` as it explores actions to account for uncertainty in the reward model. It is, however, more robust to variance in the time-dependent latent variable (see Figure 10a in the Appendix). We also observe a similar behavior for `FPG` as it performed better than `CPG` for out-of-distribution instances in Figure 5. In Figure 7, we show the results of a set of experiments to test this behavior by adding gradual noise to $X_t$ and thus decreasing the quality of the fitted LVM (see Appendix E.1 for details). We noted that `FPG-TS` gradually outperforms `FPG` with increasing noise while `CPG` was biased throughout.

**Regression modeling is sensitive to limitations of observational data** The regression baseline (greedy with respect to a time-series prediction of the reward for each action) performs poorly in the Synthetic environment (Figure 4), with a substantially larger bias than the LVM-based alternatives. This is consistent with the model fitting results at test time Appendix I and likely due to the regression model not exploiting the structure in the data like the LVMs. On ADCB, Regression initially performs well, but quickly deteriorates. This is likely because the observational data is limited to sequences of length $T_o = 200$ but test instances have much longer horizons ($T = 2000$) and the time-series model fails to extrapolate. For the Synthetic environment, we also run out-of-distribution experiments in which we move the mean of $U$ at inference time by $\Delta z = 1, \Delta z = 2, \Delta z = 4$, to the point that there is no overlap between the distributions. In Figure 5, we see that `CPG` and `FPG` are more robust to shifts in the latent variable than regression (also see Appendix E.2).

**Modeling assumptions & `ILB`** We see that `ILB`-based algorithms can be sensitive to noise in the latent state when comparing the ADCB results with different levels of latent noise ($\sigma^2 = 0.02$ in Figure 3, $\sigma^2 = 0.1$ in Figure 10a). Adaptive `FPG` and `FPG-TS` algorithms can overcome this bias to some extent, but are not as robust as MAB which remains unaffected. In a similar set of experiments, we added incremental noise to the context variable in the Synthetic setting (see Appendix E.1). In this case, oracle models performed increasingly poorly, while `FPG` and especially the `FPG-TS` algorithm could adapt their latent state estimates.

**When is online learning necessary?** The MAB baseline consistently achieves low regret by the end of exploration, as expected, but converges substantially slower than all methods based on latent variable models, including `FPG-TS`. For the Synthetic environment, the absence of bias for MAB may be sufficient for

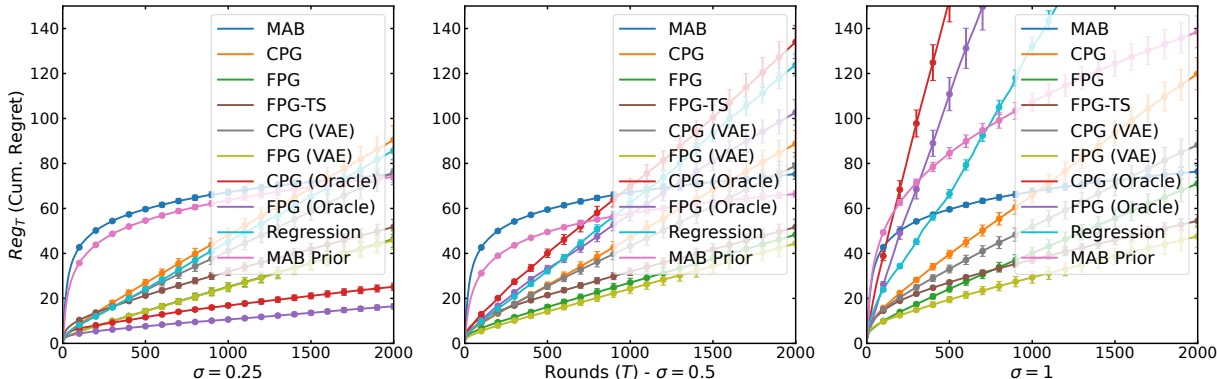

Figure 7: Expected cumulative regret `ILB` and baseline algorithms for different levels of standard deviation $\sigma = 0.25, 0.5$, and 1 Gaussian noise in the context $X_t$. LVMs are refitted for each level of noise. The error bars show standard error calculated across 1000 seeds.

it to be preferred over the VAE and Regression baselines. When the noise in the latent state is too great, such as for the uniform distribution in Figure 4, MAB becomes preferable to hybrid alternatives. We see a similar pattern on ADCB: when the latent noise increases, MAB becomes preferable over other algorithms (see Appendix E.3). This confirms the bias-variance tradeoff: a poorly fit latent variable model may find a near-optimal action quickly but suffer compared to an exploration-based algorithm in the long run.

**Offline variance and online bias** In Figure 6, we show the performance of LVM based `CPG` and `FPG` models compared to oracle based `CPG` and `FPG` and a Thompson sampling based MAB from experiments with different numbers of actions, going from $K = 10$ to $K = 50$, while keeping a fixed sample size. MAB converges each time with slower convergence time to oracle and LVM based models. Oracle-based models always converge to the best arm but need longer time for convergence due to the increasing difficulty of distinguishing the best arm. LVM based models on the other hand start to show a bias with increasing number of arms. This is an example of variance in the training time contributing to a bias in inference time.

## 5 Conclusion

In this work, we present the first provably identifiable latent bandit model learned from observational data for sample-efficient sequential decision-making. Our analysis proves a new identifiability result for a variant of nonlinear independent component analysis where latent states differ only in their mean. We investigate the conditions favorable to learn such a model and test the sensitivity of our assumptions in a semi-synthetic decision-making environment. Our theoretical and empirical results demonstrate the promise of leveraging observational data in personalized decision making.

This is a first and exploratory work investigating the conditions under which the LVM can be learned from data. Key limitations of our work are the stationarity and invertibility assumptions we need for identifiability. For future work, we plan to generalize our identifiability assumptions and model the disease progression as a time-dependent latent variable to allow for conditions changing over time. Another direction is to focus on the case where the learned LVM does not generalize well to new instances. An approach toward this goal would be to use a meta-algorithm that detects model misspecification and switches algorithms.

### Broader Impact Statement

This paper presents work whose goal is to advance decision-making through machine learning. Any application of automated decision-making must be made with caution and sufficient guard rails appropriate for the specific problem. Our work is primarily methodological and does not have direct practical implications on healthcare or other domains.

**Acknowledgments**

This work was partially supported by the Wallenberg AI, Autonomous Systems and Software Program (WASP) funded by the Knut and Alice Wallenberg Foundation.

The computations and data handling were enabled by resources provided by the National Academic Infrastructure for Supercomputing in Sweden (NAISS), partially funded by the Swedish Research Council through grant agreement no. 2022-06725.

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

# A  Notation

Table 2: Notation. Indices that indicate problem instances $i$ and time points $t$ are dropped when clear from context (e.g., when stated to be fixed in text or in i.i.d. distributions over multiple instances).

| Random variables | |
|---|---|
| $Z_i$ | Latent state for problem instance $i$ |
| $Z_{i,t}$ | Time-varying (noisy) latent state for problem instance $i$ at time $t$ |
| $U$ | Population distribution of latents $Z_i$ |
| $X_{i,t}$ | Context for instance $i$ at time $t$ |
| $A_{i,t}$ | Action for instance $i$ at time $t$ |
| $R_{i,t}$ | Reward for instance $i$ at time $t$ |
| $\eta_{i,t}$ | Noise variable for $Z_{i,t}$ |
| $\epsilon_{i,t}$ | Noise variable for $R_{i,t}$ |
| $H_{i,t}$ | Stochastic history of past contexts, actions and rewards up to time $t$ for instance $i$ |
| $\mathrm{Reg}_T$ | Expected cumulative regret |
| **Observations and constants** | |
| $\mathcal{D}$ | Observational dataset of triplets $(x_{i,t}, a_{i,t}, r_{i,t})$ collected across instances and time |
| $z_i$ | Latent state for problem instance $i$ |
| $z_{i,t}$ | Time-varying (noisy) latent state for problem instance $i$ at time $t$ |
| $x_{i,t}$ | Context for instance $i$ at time $t$ |
| $a_{i,t}$ | Action for instance $i$ at time $t$ |
| $r_{i,t}$ | Reward for instance $i$ at time $t$ |
| $T_i$ | Number of observations for instance $i$ in the dataset $\mathcal{D}$ |
| $I$ | Total number of instances included in the dataset $\mathcal{D}$ |
| $n$ | Dimension of the latent state |
| $d$ | Dimension of the context |
| $\mathcal{A}$ | Set of all actions |
| $K$ | Total number of actions (i.e. $|\mathcal{A}|$) |
| $\mu_a(z)$ | Expected reward for action $a$ conditioned on latent state $z$ |
| $\sigma^2$ | Variance associated with the latent noise variable $\eta_{i,t}$ |
| $\mu_i^*$ | Optimal reward for instance $i$ |
| $a_i^*$ | Optimal action for instance $i$ |
| $M$ | Matrix of all instance means $z_i$ |
| $B, b$ | Affine transformation constants for the identification of latent state |
| **Functions** | |
| $g$ | Non-linear transformation mapping the latent state to context variable |
| $f$ | Feature extractor for the learned representation from context to latent variables |
| $W, b$ | Weight and biases for the multinomial linear regression |
| $\theta_a$ | Reward function for action $a$ |

# B  Identifiability of the latent variable model

In Section 2, we assume that the reward is generated according to a *latent variable* $Z_i$, and propose to make use of the observational historical data $\mathcal{D}$ to learn an LVM $\hat{p}(Z|H_t, X_t)$, recovering $Z_i$. However, as $Z_i$ is not observed different learning algorithms may fit the observed data $\mathcal{D}$ equally well yet differ in their estimates $\hat{p}(Z|H_t, X_t)$, which potentially leads to different estimates for the optimal action. This raises two questions: (i) How do we define the notion that $Z_i$ is uniquely recoverable from the distribution of observable data $p(H_T)$? (ii) What kind of learning algorithm provably recovers $p(Z|H_t, X_t)$? Here we only answer the first question (i) and give a definition of *identifiability* in Definitions B.1 and B.2, inspired by related works in the literature Basse & Bojinov (2020); Khemakhem et al. (2020a). We answer the latter question (ii) in Section 3 and Appendix B.1.

**Definition B.1** (Identifiability of LVM). We say that the latent variable model $p(Z|H_T, X_T)$ is identifiable from the distribution of the observed data $p(H_T, X_T)$, with respect to family $\mathcal{F}$ if for any $f, f' \in \mathcal{F}$:

$$\forall f, f' : \ \hat{p}_f(Z|H_T, X_T) = p(Z|H_T, X_T) = \hat{p}_{f'}(Z|H_T, X_T) \implies f = f'. \tag{9}$$

One difficulty with Definition B.1 is that the model family $\mathcal{F}$ needs to be just expressive enough to uniquely capture any $g$. However, when $g$ is unobserved and nonparametric as in Assumption 3.1 finding a good candidate model family becomes difficult. Instead one can try to achieve a *partial identifiability* of $g$ where the function is recovered up to a class of functions defined by some equivalence relation. An example equivalence relation is the invertible affine transformation we use in Theorem 3.3.

**Definition B.2** (Affine Identifiability). We define affine equivalence relation $\sim$ on $\mathcal{F}$ as

$$f \sim f' \iff \exists A, a : f = Af' + a \tag{10}$$

for vector $a$ and an invertible matrix $A$. We say that the LVM $p(Z|H_T, X_T)$ is affine identifiable if

$$\forall f, f' : \ \hat{p}_f(Z|H_T, X_T) = p(Z|H_T, X_T) = \hat{p}_{f'}(Z|H_T, X_T) \implies f \sim f'. \tag{11}$$

## B.1  Proof of Theorem 3.3

Hyvärinen and Morioka (Hyvarinen & Morioka, 2016) give an argument for recovering the conditional probability of the patient/instance indicator (in their case, "segment"), stated here as Lemma B.3.

**Lemma B.3** ((Hyvarinen & Morioka, 2016)). *For the classifier given in equation (4), in the limit of per-instance infinite data the optimal feature extractor $f^\star$ given by (5) would converge to the true posterior $p(C|X)$:*

$$p\left(C = c \mid X = x\right) = \frac{p_c\left(X = x\right) p\left(C = c\right)}{\sum_{j=1}^{I} p_j\left(X = x\right) p\left(C = j\right)}, \tag{12}$$

*where $C$ is the (instance) class label of $X$, $p_c(X = x) = p(X = x | C = c)$ is the conditional distribution of the context for instance class $c$, and $p(C = c)$ are prior distributions for each instance. Then we have for $f^\star(x)$:*

$$W_c^T f^\star(x) + b_c = \log p_c(x) - \log p_1(x) + \rho_c, \tag{13}$$

*where $\rho_c = \log \frac{p(C=c)}{p(C=1)}$ relates to the length (number of samples) of each instance sequence.*

**Theorem 3.3 (Restated).** *Under Assumptions 3.1–3.2 the optimal feature extractor $f^\star$, according to (5), is equal to the inverse emission function $g^{-1}$ up to an invertible affine transformation. In other words,*

$$Bf^\star(x) + b = g^{-1}(x)$$

*for constant invertible matrix $B \in \mathbb{R}^{n \times n}$, and $b \in \mathbb{R}^n$.*

*Proof.* According to Assumption 3.1 the conditional distribution of each $Z_{i,t}$ will be normally distributed around the true mean $Z_i$, with mean $z_i$ and variance $\sigma^2$. For each time point $z_{i,.}$ the log-pdf of the product distribution can be written as:

$$\log \mathbb{P}(Z_{i,.} = \zeta) = \log p_i(\zeta) = \sum_{j=1}^{n} \frac{(\zeta_j - z_{i,j})^2}{\sigma^2}, \tag{14}$$

where we use $j$ to indicate the dimension. Since $g$ is injective we can not use the change of variables formula directly, instead we use a generalization for injective functions Kothari et al. (2021); Federer (2014):

$$\log p_X(x) = \log p_Z(g^{-1}(x)) - \frac{1}{2} \log \left| \det \left[ \mathbf{J}_g(g^{-1}(x))^\top \mathbf{J}_g(g^{-1}(x)) \right] \right|$$

where $\mathbf{J}$ is the Jacobian matrix. For our case, this yields the equation:

$$\log p_i(x) = \sum_{j=1}^{n} \frac{(g_j^{-1}(x) - z_{i,j})^2}{\sigma^2} - \frac{1}{2} \log \left| \det \left[ \mathbf{J}_g(g^{-1}(x))^\top \mathbf{J}_g(g^{-1}(x)) \right] \right|, \tag{15}$$

We look at the instance with index $i = 1$, following from line (15), we have:

$$\log p_1(x) = \sum_{j=1}^{n} \frac{(g_j^{-1}(x) - z_{1,j})^2}{\sigma^2} - \frac{1}{2} \log \left| \det \left[ \mathbf{J}_g(g^{-1}(x))^\top \mathbf{J}_g(g^{-1}(x)) \right] \right| \tag{16}$$

Using (16) for the $\log p_1$ term in Lemma B.3:

$$\log p_i(x) = \sum_{j=1}^{n} \left[ W_{i,j} f_j^\star(x) + \frac{(g_j^{-1}(x) - z_{1,j})^2}{\sigma^2} \right] + b_i - \rho_i - \frac{1}{2} \log \left| \det \left[ \mathbf{J}_g(g^{-1}(x))^\top \mathbf{J}_g(g^{-1}(x)) \right] \right| \tag{17}$$

Finally, taking (17) and (15) equal for arbitrary $i$, the Jacobian terms cancel:

$$\sum_{j=1}^{n} \frac{(g_j^{-1}(x) - z_{i,j})^2 - (g_j^{-1}(x) - z_{1,j})^2}{\sigma^2} = \sum_{j=1}^{n} W_{i,j} f_j^\star(x) + b_i - \rho_i. \tag{18}$$

After canceling the $(g_j^{-1}(x))^2$ terms in (18) we get

$$\sum_{j=1}^{n} \frac{2g_j^{-1}(x)(z_{1,j} - z_{i,j}) + z_{i,j}^2 - z_{1,j}^2}{\sigma^2} = \sum_{j=1}^{n} W_{i,j} f_j^\star(x) + b_i - \rho_i,$$

and simplify for $\mathbf{b}_i = b_i - \rho_i - \sum_j \frac{z_{i,j}^2 - z_{1,j}^2}{\sigma^2}$ and $\mathbf{B}_{i,j} = \frac{2(z_{1,j} - z_{i,j})}{\sigma^2}$ which yields

$$\sum_{j=1}^{n} \mathbf{B}_{i,j} g_j^{-1}(x) = \sum_{j=1}^{n} W_{i,j} f_j^\star(x) + \mathbf{b}_i. \tag{19}$$

The equation (19) can be written in the matrix form as

$$\mathbf{B} g^{-1}(x) = \mathbf{W} f^\star(x) + \mathbf{b} \tag{20}$$

where we collect the entries of $\mathbf{b}_i$ in vector $\mathbf{b}$, $W_{i,j}$ in the weight matrix $\mathbf{W}$, and $\mathbf{B}_{ij}$ in the matrix $\mathbf{B}$ for all $I$ instances. When means $z_i$ are sufficiently different, in particular when there are at least n linearly independent components as per Assumption 3.2 (b) then $\mathbf{B}$ is full rank which implies that the pseudo inverse satisfies $\mathbf{B}^\dagger \mathbf{B} = \mathbf{I}$. Multiplying both sides by $\mathbf{B}^\dagger$ gives the desired result. $\qquad\square$

## C Identifiability of decision-making criteria

**Theorem 3.4 (Restated).** *Assume that reward means are linear, $\mu_a(z) = \theta_a^\top z$, and fix a problem instance $i$. Then, under the conditions of Assumption 3.1, the state-conditional expected reward $\mathbb{E}[R_{i,t} \mid Z_i = z, \mathbf{do}(A_{i,t} = a)]$ of an intervention $a$ is identifiable from the observational distribution of problem instances $p(H_T)$ by the OLS regression estimand applied to observed rewards and latent states inferred by an optimal feature extractor $f$ in the sense of Theorem 3.3.*

**Proof.** First, let's begin with the identification of $\mathbb{E}[R_{i,t} \mid Z_i, \mathbf{do}(a)]$ under the assumption that $Z$ could be observed directly and generalize this later. Under Assumption 3.1, the reward is stationary conditioned on $Z$ and the action.

### Step 1: Causal identifiability of the reward model

Assume that the system of variables $Z_i, Z_{i,t}, X_{i,t}, A_{i,t}, R_{i,t}$ for all instances $i$ and time points $t$ obey the structural causal model of Assumption 3.1. Then, for a fixed instance $i$, at all time points $t$, the causal graph in our structural causal model satisfies the backdoor criterion (Pearl, 2009) for the effect on $R_{i,t}$ of an intervention on $A_{i,t}$ by conditioning on $Z$. In other words, $Z$ blocks all backdoor paths from $R_{i,t}$ ending in $A_{i,t}$. Therefore,

$$\mathbb{E}[R_{i,t} \mid Z_i = z, \mathbf{do}(A_{i,t} = a)] = \mathbb{E}[R_{i,t} \mid Z_i = z, A_{i,t} = a].$$

Moreover, since $R_{i,t}$ is stationary in both time and across problem instances conditioned on $a$ and $z$,

$$\mathbb{E}[R_{i,t} \mid Z_i = z, A_{i,t} = a] = \mathbb{E}[R \mid Z = z, A = a] \ .$$

Hence, the expected reward following an intervention $a$ is identifiable from the observational distribution $p(X_1, A_1, R_1, ..., X_T, A_T, R_T)$[1] under the data-generating process of Assumption 3.1.

### Step 2: Identification without observing $Z$

From Step 1, it is clear that we can identify the expected reward of an action conditioned on the fixed latent state $Z_i$ of an individual. However, since the latent state is unobserved, we must infer it from observed variables for the reward to be identifiable. First, assume that we have access to the oracle LVM $(g^{-1}, \theta)$ that generated the observational data and the current problem instance. We will generalize this to invariance under an affine transform later.

For any time step $t$ and instance $i$, it holds under Assumption 3.1 that

$$Z_{i,t} = g^{-1}(X_{i,t}) \quad \text{and} \quad \mathbb{E}[R_{i,t} \mid Z_i, A_{i,t} = a] = \theta_a^\top Z_i \ .$$

Moreover, since $\forall t : Z_{i,t} \sim \mathcal{N}(Z_i, \sigma^2 \mathbb{I})$ by assumption, $\mathbb{E}[Z_{i,t} \mid Z_i] = Z_i$. Since $Z_{i,t}$ is stationary in time given $Z_i$, we may drop the time index and view this expectation as an integral in time. Due to injectivity, we have for the left-inverse of $g^{-1}$

$$\mathbb{E}[R_{i,t} \mid Z_i, A_{i,t} = a] = \mathbb{E}[R_{i,t} \mid \mathbb{E}[g^{-1}(X_{i,\cdot})], A_{i,t} = a] = \theta_a^\top \mathbb{E}[g^{-1}(X_{i,\cdot})] \ .$$

From Theorem 3.3, we know that $g^{-1}$ can be identified up to an affine transformation. We'll deal with this invariance next.

### Step 3: Invariance to affine transform

Since $Z_i$ is not observed directly, we rely on the learned representation $\hat{Z}_{i,t} = f(X_{i,t})$. Dropping the instance index $i$, by Theorem 3.3, a feature extractor $f$ may be partially identified from the observational distribution such that $\hat{z}_t = f(x_t)$ satisfies:

$$z_t = B\hat{z}_t + b,$$

---

[1] We suppress the instance index $i$ since instances are assumed to be i.i.d.

where $B$ is an invertible matrix, and $b$ is a constant vector. Substituting $Z$ in terms of $\hat{Z}$ into the reward model for a fixed instance $i$, following action $A_t = a$ and dropping the time index for convenience,

$$R = \theta_a^\top Z + \epsilon_a = \theta_a^\top \mathbb{E}[Z_t] + \epsilon_A = \theta_a^\top \mathbb{E}[B\hat{z}_{(.)} + b] + \epsilon_a = \theta_a^\top (B\mathbb{E}[\hat{z}_{(.)}] + b) + \epsilon_a.$$

Introducing transformed coefficients: $\tilde{\theta}_a = B\theta_a$ and $\tilde{b}_a = \theta_a^\top b$, we find that

$$R = \tilde{\theta}_a^\top \mathbb{E}[\hat{z}_{(.)}] + \tilde{b}_a + \epsilon_a.$$

Thus, the expected reward depends linearly on $\hat{z} = \mathbb{E}[\hat{z}_{(.)}]$, with transformed coefficients,

$$\mathbb{E}[R \mid Z = z, \mathbf{do}(A = a)] = \tilde{\theta}_a^\top \hat{z} + \tilde{b}_a .$$

Now, consider a dataset generated by inferring $Z$ from the observational distribution $p(H_T)$ with samples $(\hat{z}_i, a_{i,t}, r_{i,t})$ for a range of instances $i$ and time points $t$, where $\hat{z}_i = \mathbb{E}[f(x_{i,.})]$. The ordinary least squares (OLS) estimator applied separately to samples sets $\{(\hat{z}_i, r_{i,t})\}$ for each action will return parameters $(\tilde{\theta}_a, \tilde{b}_a)$ in expectation, since OLS is unbiased. Hence, $\mathbb{E}[R \mid Z, A = a]$ is identifiable from the observational distribution. $\qquad\square$

## D $\ $ CPG **has constant regret**

In this section, we go over the proof of Theorem 3.5. We assume that we have access to an optimal feature extractor $f$ in the sense of Theorem 3.3 and an OLS estimate of the rewards $\hat{\theta}$ as in Theorem 3.4, from these two we develop a notion of optimal model pair $(\hat{\theta}, f)$ where $\hat{\theta}^\top f(x) = \theta^\top g^{-1}(x)$ for $x \in \mathbb{R}^d$. It follows that for the optimal model pair we have

$$\theta^\top g^{-1}(x) = \theta^\top (Bf(x) + b) = \theta^\top Bf(x) + \theta^\top b = \hat{\theta}^\top f(x) + \hat{\theta}_0, \tag{21}$$

where $\hat{\theta}_0$ is the fitted intercept term. The equation (21) yields the following relationship between $\hat{\theta}$ and the true $\theta$:

$$\hat{\theta} = \theta^\top B \tag{22}$$

$$\hat{\theta}_0 = \theta^\top b \tag{23}$$

We use this relationship in the following Lemma D.1 where we show that FPG estimate is unbiased and follows a Gaussian distribution centered around $f(g(z_i))$.

**Lemma D.1** (Estimator for FPG)**.** *For an instance $i$, with latent state $z_i$, under an optimal model pair $(\hat{\theta}, f)$, in the sense of Remark 1, the estimate for FPG, $\hat{z}_i$, as given in equation (7) in Algorithm 1, for any time step $t$, is distributed Gaussian around $f(g(z_i))$, the fixed affine transform around the true instance mean, $z_i$.*

*Proof.* The FPG algorithm decides on $\hat{z}_i$ for each time point by $t$ using the convex optimization problem (7) applied to the current history and context, $h_t, x_t$, in Algorithm 1. We write the corresponding loss function for the optimization problem as $\ell(z)$:

$$\ell(z) = \sum_{t'=1}^{t} (r_{t'} - \hat{\theta}_{a_{t'}}^\top z - \hat{\theta}_{0,a_{t'}})^2 + \|z - \bar{z}_t\|^2$$

where we used the short hand $\bar{z}_t = \frac{1}{t}\sum_{t'=1}^{t} f(x_{t'})$ for the LVM mean estimate. Taking the gradient with respect to $z$ gives

$$-\frac{1}{2}\nabla_z \ell = \sum_{t'=1}^{t} \hat{\theta}_{a_{t'}} ((r_{t'} - \hat{\theta}_{0,a_{t'}}) - \hat{\theta}_{a_{t'}}^\top z) + (\bar{z}_t - z),$$

which can be rewritten by rearranging terms as:

$$= \sum_{t'=1}^{t} \hat{\theta}_{a_{t'}}(r_{t'} - \hat{\theta}_{0,a_{t'}}) + \bar{z}_t - \left( \mathbb{I} + \sum_{t'=1}^{t} \hat{\theta}_{a_{t'}} \hat{\theta}_{a_{t'}}^{\top} \right) z.$$

Taking $-\frac{1}{2}\nabla_z \ell = 0$ and moving $z$ term to the left-hand side gives the estimate:

$$\hat{z}_i = \left( \mathbb{I} + \sum_{t'=1}^{t} \hat{\theta}_{a_{t'}} \hat{\theta}_{a_{t'}}^{\top} \right)^{-1} \left[ \bar{z}_t + \sum_{t'} \hat{\theta}_{a_{t'}}(r_{t'} - \hat{\theta}_{0,a_{t'}}) \right], \tag{24}$$

which gives us a closed form for $\hat{z}_i$ that minimizes $\ell(z)$. Next, we wish to show that $\hat{z}_i$ is normally distributed around $f(g(z_i))$. By the affine identifiability presented in Theorem 3.3, we have :

$$f(x_t) = f(g(z_i + \eta_t)) = B^{-1}(g^{-1}(g(z_i + \eta_t)) - b) = f(g(z_i)) + \tilde{\eta}_t \tag{25}$$

$$r_t = \theta_{a_t}^{\top} z_i + \epsilon_{a_t} = \hat{\theta}_{a_t}^{\top} f(g(z_i)) + \hat{\theta}_{0,a_t} + \epsilon_{a_t} \tag{26}$$

where $z_i$ is the true instance mean, $\tilde{\eta}_t = B^{-1}\eta_t$, and $\hat{\theta}_{0,a} = \theta_a^{\top} b$ results from affine identifiability. We discuss in the Remark on Remark 1 the effect of affine identifiability on the rewards. In practice, for an optimal model pair the fitted OLS estimator $\hat{\theta}$ will have an intercept term $\hat{\theta}_0$ as given by (23). Applying (25) and (26) to (24) we get

$$\hat{z}_i = \left( \mathbb{I} + \sum_{t'=1}^{t} \hat{\theta}_{a_{t'}} \hat{\theta}_{a_{t'}}^{\top} \right)^{-1} \left[ \bar{z}_t + \sum_{t'} \hat{\theta}_{a_{t'}}(r_{t'} - \hat{\theta}_{0,a_t'}) \right] \tag{27}$$

$$= \left( \mathbb{I} + \sum_{t'=1}^{t} \hat{\theta}_{a_{t'}} \hat{\theta}_{a_{t'}} \right)^{-1} \left[ f(g(z_i)) + \frac{1}{t} \sum_{t'=1}^{t} \tilde{\eta}_{t'} + \sum_{t'} \hat{\theta}_{a_{t'}}(r_{t'} - \hat{\theta}_{0,a_t'}) \right] \tag{28}$$

$$= \left( \mathbb{I} + \sum_{t'=1}^{t} \hat{\theta}_{a_{t'}} \hat{\theta}_{a_{t'}}^{\top} \right)^{-1} \left[ f(g(z_i)) + \frac{1}{t} \sum_{t'=1}^{t} \tilde{\eta}_{t'} + \sum_{t'} \hat{\theta}_{a_{t'}}(\hat{\theta}_{a_t}^{\top} f(g(z_i)) + \hat{\theta}_{0,a_t'} + \epsilon_{a_{t'}} - \hat{\theta}_{0,a_t'}) \right], \tag{29}$$

Equation (29) can be broken into deterministic and stochastic parts which simplifies to

$$\hat{z}_i = \left( \mathbb{I} + \sum_{t'=1}^{t} \hat{\theta}_{a_{t'}} \hat{\theta}_{a_{t'}}^{\top} \right)^{-1} \left[ f(g(z_i)) + f(g(z_i)) \sum_{t'} \hat{\theta}_{a_{t'}} \hat{\theta}_{a_{t'}}^{\top} \right] \tag{30}$$

$$+ \left( \mathbb{I} + \sum_{t'=1}^{t} \hat{\theta}_{a_{t'}} \hat{\theta}_{a_{t'}}^{\top} \right)^{-1} \left[ \frac{1}{t} \sum_{t'=1}^{t} \tilde{\eta}_{t'} + \sum_{t'} \epsilon_{a_{t'}} \hat{\theta}_{a_{t'}} \right] \tag{31}$$

$$= f(g(z_i)) + \left( \mathbb{I} + \sum_{t'=1}^{t} \hat{\theta}_{a_{t'}} \hat{\theta}_{a_{t'}}^{\top} \right)^{-1} \left[ \frac{1}{t} \sum_{t'=1}^{t} \tilde{\eta}_{t'} + \sum_{t'} \epsilon_{a_{t'}} \hat{\theta}_{a_{t'}} \right] \tag{32}$$

As $\eta_t$ and $\epsilon_{a_t'}$ are mean-zero Gaussian the stochastic part of (32) is a linear combination of mean-zero Gaussians. This concludes the proof that the estimate for FPG is Gaussian around $f(g(z_i))$, the fixed affine transform around the true mean, $z_i$. $\square$

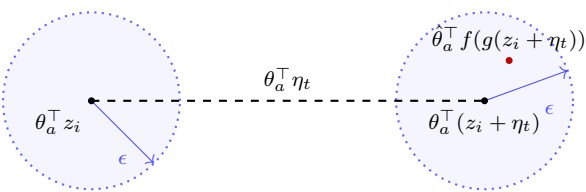

Figure 8: The bias in the `CPG` reward means estimate, under a uniform bound on the error of $\hat{\theta}^\top f(x_t)$.

Now, we show our result on cumulative regret of the `CPG` algorithm. It follows from standard concentration results for sub-Gaussian random variables, see Theorem D.2.

**Theorem 3.5 (Restated).** *For an instance $i$, let $\Delta_i > 0$ such that $\forall a \neq a^* : |(\theta_{a^*} - \theta_a)^\top z_i| > \Delta_i$, and let $\overline{\Delta}_i = \max_{a \neq a^*}(\theta_{a^*} - \theta_a)^\top z_i$, and assume w.l.o.g. that $\forall a : \|\theta_a\|_2 = 1$. Then for a learned model pair $(f, \hat{\theta})$ such that $\forall a, z : \left|\hat{\theta}^\top f(g(z)) - \theta^\top z\right| < \epsilon$, the expected regret of `CPG` is bounded by*

$$\text{Reg}_T \leq \mathbb{1}_{[\epsilon < \frac{\Delta_i}{2}]}\frac{8K\sigma^2\overline{\Delta}_i}{(\Delta_i - 2\epsilon)^2} + \mathbb{1}_{[\epsilon \geq \frac{\Delta_i}{2}]}\left(\frac{8K\sigma^2\overline{\Delta}_i}{\Delta_i^2} + 2\epsilon T + \frac{\Delta_i T}{2}\right)$$

*where $\sigma^2$ is the variance of $\eta_{i,t}$ (of $Z_{i,t}$ given $Z_i$).*

*Proof.* We start by looking at the conditions under which `CPG` can play the optimal arm, then we will quantify the regret in each case.

**Step 1: Conditions for playing the optimal arm.** `CPG` will play a sub-optimal arm $a$ if $\hat{\theta}_a^\top \hat{z}_{i,t} \geq \hat{\theta}_{a^*}^\top \hat{z}_{i,t}$ where

$$\hat{z}_t := \frac{1}{t}\sum_{t'=1}^{t} f(x_{i,t'}),$$

for `CPG`. Hence, if for all arms we have

$$\left|\hat{\theta}_a^\top \hat{z}_t - \theta_a^\top z_i\right| \leq \frac{\Delta_i}{2}$$

`CPG` will play the optimal arm. We can write $\hat{\theta}_a^\top \hat{z}_t - \theta_a^\top z_i$ as a sum of the terms

$$\hat{\theta}_a^\top \hat{z}_t - \theta_a^\top z_i = \left(\frac{1}{t}\sum_{t'=1}^{t}\hat{\theta}_a^\top f(x_{i,t'}) - \frac{1}{t}\sum_{t'=1}^{t}\theta_a^\top z_{i,t'}\right) + \left(\frac{1}{t}\sum_{t'=1}^{t}\theta_a^\top z_{i,t'} - \theta_a^\top z_i\right).$$

In Figure 8, we show how the respective summands are bounded either by $\epsilon$ or by the Gaussian concentration, using triangle inequality we get

$$\left|\hat{\theta}_a^\top \hat{z}_t - \theta_a^\top z_i\right| \leq \underbrace{\left|\frac{1}{t}\sum_{t'=1}^{t}[\hat{\theta}_a^\top f(g(z_{i,t'})) - \theta_a^\top z_{i,t'}]\right|}_{\leq \epsilon} + \left|\frac{1}{t}\sum_{t'=1}^{t}\theta_a^\top z_{i,t'} - \theta_a^\top z_i\right|$$

$$\leq \epsilon + \frac{1}{t}\left|\sum_{t'=1}^{t}\theta_a^\top \eta_{t'}\right|$$

where each element in $\eta_{t'}$ is $\mathcal{N}(0, \sigma^2)$. Thus `CPG` plays the optimal arm when

$$\frac{1}{t}\left|\sum_{t'=1}^{t}\theta_a^\top \eta_{t'}\right| < \frac{\Delta_i - 2\epsilon}{2}. \tag{33}$$

However, when $\Delta_i \leq 2\epsilon$ this term is never satisfied and `CPG` suffers linear regret. We treat the case $\Delta_i < 2\epsilon$ first and return to it later.

**Step 2: Constant regret when the error is small.** When $\Delta_i > 2\epsilon$, we can characterize the regret, based on the concentration of the $\frac{1}{t}\left|\sum_{t'} \theta_a^\top \eta_{t'}\right|$ term in (33), where $\eta_t \sim \mathcal{N}(0, \sigma^2)$. Recall that the regret is given by

$$\text{Reg}_T = \mathbb{E}\left[\sum_{t=1}^T (\mu_i^* - R_{i,t})\right] = \sum_{t=1}^T \mathbb{E}[z_i^\top(\theta_{a^*} - \theta_{A_t})]$$

The regret increases by selecting a suboptimal action, which happens whenever the noisy estimate $\hat{z}_t$ ranks a suboptimal action over the optimal one. Using the union bound over probability of suboptimal selections,

$$\text{Reg}_T \leq \sum_{t=1}^T \sum_a \overline{\Delta}_i P\left(\left|\theta_a^\top \sum_{t'=1}^t \eta_{t'}\right| \geq \frac{\Delta_i - 2\epsilon}{2}\right) \tag{34}$$

where $\overline{\Delta}_i = \max_{a \neq a^*} z_i^\top(\theta_{a^*} - \theta_a)$. We now apply Theorem D.2 with $w = \theta_a$ which yields

$$P\left(\left|\theta_a^\top \sum_{t'=1}^t \eta_{t'}\right| \geq \frac{\Delta_i - 2\epsilon}{2}\right) \leq 2\exp\left[-\frac{t(\Delta_i - 2\epsilon)^2}{4\sigma^2}\right]$$

since $\|\theta_a\| = 1$. Putting it together yields

$$\text{Reg}_T \leq K\overline{\Delta}_i \sum_{t=1}^\infty 2\exp\left[-\frac{t(\Delta_i - 2\epsilon)^2}{4\sigma^2}\right] = 2K\overline{\Delta}_i \left(\exp\left[\frac{(\Delta_i - 2\epsilon)^2}{4\sigma^2}\right] - 1\right)^{-1}.$$

Now using the fact that

$$\frac{1}{e^x - 1} \leq \frac{1}{x}$$

yields

$$\text{Reg}_T \leq \frac{8K\sigma^2\overline{\Delta}_i}{(\Delta_i - 2\epsilon)^2}. \tag{35}$$

**Step 3. Linear regret when the model error is large.** When $2\epsilon \geq \Delta_i$, `CPG` can not distinguish between the arms in the set

$$\tilde{A}_{\min} := \{a \in \mathcal{A} : (\theta_{a^*} - \theta_a)^\top z_i \leq 2\epsilon\},$$

we call $\tilde{A}_{\min}$ the minimal set of arms. At each round `CPG` plays an arm from the set

$$\tilde{A}_t := \left\{a \in \mathcal{A} : (\theta_{a^*} - \theta_a)^\top z_i \leq 2\epsilon + \frac{1}{t}\left|\sum_{t'=1}^t \theta^\top \eta_{t'}\right|\right\}$$

which will reduce to the minimal set of arms when $\left|\frac{1}{t}\sum_{t'=1}^t \theta^\top \eta_{t'}\right| < \frac{\delta}{2}$, for $\delta > 0$, as the next best arm could be arbitrarily close. One can simply select any value for $\delta$ with an additional linear penalty. Selecting $\delta = \frac{\Delta_i}{2}$ leads to a regret at each round

$$\text{Reg}_T \leq 2\epsilon T + \frac{\Delta_i T}{2} + \sum_{t=1}^T \sum_a \overline{\Delta}_i P\left(\left|\theta_a^\top \sum_{t'=1}^t \eta_{t'}\right| \geq \frac{\Delta_i}{2}\right) \tag{36}$$

Notice the similarity of terms between (34) and (36). We proceed with the same steps replacing $\frac{\Delta - 2\epsilon}{2}$ with $\frac{\Delta_i}{2}$ which results in

$$\text{Reg}_T \leq \frac{8K\sigma^2\overline{\Delta}_i}{\Delta_i^2} + 2\epsilon T + \frac{\Delta_i T}{2} \tag{37}$$

**Step 4. Combining the two cases.** Combining the two cases (35) and (37) gives the bound

$$\text{Reg}_T \leq \mathbb{1}_{[\epsilon < \frac{\Delta_i}{2}]} \frac{8K\sigma^2\overline{\Delta}_i}{(\Delta_i - 2\epsilon)^2} + \mathbb{1}_{[\epsilon \geq \frac{\Delta_i}{2}]} \left( \frac{8K\sigma^2\overline{\Delta}_i}{\Delta_i^2} + 2\epsilon T + \frac{\Delta_i T}{2} \right). \tag{38}$$

$\square$

**Theorem D.2** (General Hoeffding's Inequality (Vershynin, 2018))**.** *Let $X_1, ..., X_d$ be independent, zero-mean, sub-Gaussian random variables and let $w \in \mathbb{R}^d$. Then for every $\gamma > 0$*

$$P\left( |\sum_{i=1}^d X_i w_i| \geq \gamma \right) \leq 2 \exp\left[ -\frac{\gamma^2}{Q^2 ||w||_2^2} \right]$$

*with $Q^2$ equal to the maximum variance of any of the $X_i$:s.*

### Comparison of Theorem 3.5 to Latent Bandits Revisited Hong et al. (2020)

In Latent Bandits Revisited, the authors show a similar result in Theorem 2 that if the deviations in reward estimates are uniformly bounded by a constant $\epsilon$, this introduces an additional linear regret term while preserving the remaining terms of their bound. The assumption and the result of Theorem 3.5 is similar in spirit, we assume for a learned model pair $(f, \theta)$ the reward prediction error is uniformly bounded such that for all $a, z$

$$\left| \hat{\theta}^\top f(g(z)) - \theta^\top z \right| < \epsilon.$$

Under this assumption, the regret of `CPG` satisfies

$$\text{Reg}_T \leq \mathbb{1}_{[\epsilon < \frac{\Delta_i}{2}]} \frac{8K\sigma^2\overline{\Delta}_i}{(\Delta_i - 2\epsilon)^2} + \mathbb{1}_{[\epsilon \geq \frac{\Delta_i}{2}]} \left( \frac{8K\sigma^2\overline{\Delta}_i}{\Delta_i^2} + 2\epsilon T + \frac{\Delta_i T}{2} \right).$$

The regret bound in Hong et al. (2020) scales as $\sqrt{T \log T}$ and includes a similar linear $2\epsilon T$ term:

$$\text{Reg}_T \leq T\delta + 3|\mathcal{S}| + 2T\varepsilon + 2\sigma\sqrt{6|\mathcal{S}|T \log T}.$$

The remaining terms depend on the respective settings. In their setting the context variable is marginally independent of the latent state, but rewards depend directly on the context. Thus, the context alone is not useful for inferring the latent state. This is distinct from our setting where context and rewards are determined by $Z_i$ and are conditionally independent given $Z_i$. We give a summary of the settings in Table 3.

Table 3: Comparison between the `ILB` framework and the setting of Hong et al. (2020).

|  | ILB | Latent Bandits Revisited |
|---|---|---|
| Number of arms | finite | finite |
| Latent state | continuous | discrete |
| Context $X_t$ | deterministic function of latents $X_{i,t} = g(Z_{i,t})$ | independent of latents |
| Reward distribution | Gaussian ($\sigma^2$ variance) | sub-Gaussian ($\sigma^2$ proxy variance) |
| Regret bound | linear + constant | linear + square-root |

Table 4: The LVM fitting results for increasing noise in $X_t$. We fit the models for $L = 2$ and $T_o = 200$, the results are averaged over 10 seeds.

| Synthetic Data | | | |
|---|---|---|---|
| $\sigma$ | Model | $R_R^2$ | % $a^*$ |
| 0.25 | LVM | 0.73 | 76 |
| 0.25 | VAE | 0.73 | 80 |
| 0.25 | Reg. | 0.72 | 69 |
| 0.5 | LVM | 0.71 | 75 |
| 0.5 | VAE | 0.72 | 74 |
| 0.5 | Reg. | 0.70 | 61 |
| 1 | LVM | 0.70 | 68 |
| 1 | VAE | 0.73 | 77 |
| 1 | Reg. | 0.65 | 56 |

# E    Additional Experiments

## E.1    Ablation for identifiability

To evaluate the robustness of `CPG` and `FPG` to violations of our identifiability assumptions, we conduct an ablation study in which we gradually add Gaussian noise to the observed context $X_t$ in the synthetic setting. This perturbation violates the assumptions in Assumption 3.1 and introduces significant bias, particularly for oracle models that rely on the true latent structure. We first fit `ILB` and VAE-based LVMs, and regression baselines on the perturbed datasets, and then compare their performance in the online setting.

As shown in Figure 7, oracle-based models degrade substantially as the noise level increases. The regression baseline also exhibits substantial bias and reduced predictive performance. Among the learned latent-variable models, VAE-based LVMs outperform `ILB`-based models in this setting, which is consistent with their stronger model-fitting performance in Table 4.

We further observe that `FPG`, for both `ILB` and VAE-based models, adapts better to increasing noise than `CPG`, because it incorporates reward feedback when estimating the latent state. Moreover, the benefit of exploration is evident, as `FPG-TS` outperforms both greedy algorithms.

## E.2    Out of distribution experiments

In this set of experiments, we evaluate generalization to out-of-distribution instances. The results in Figure 9 show the effect of distribution shift on model performance between the LVM and a regression model for increasing difference in $\Delta z = 1, 2$, and 4. `FPG`, `CPG`, and regression models show an increase in bias while LVM-based models outperform the baseline regression in every case. `FPG` generalizes better compared to `CPG` due to using the signal in the reward.

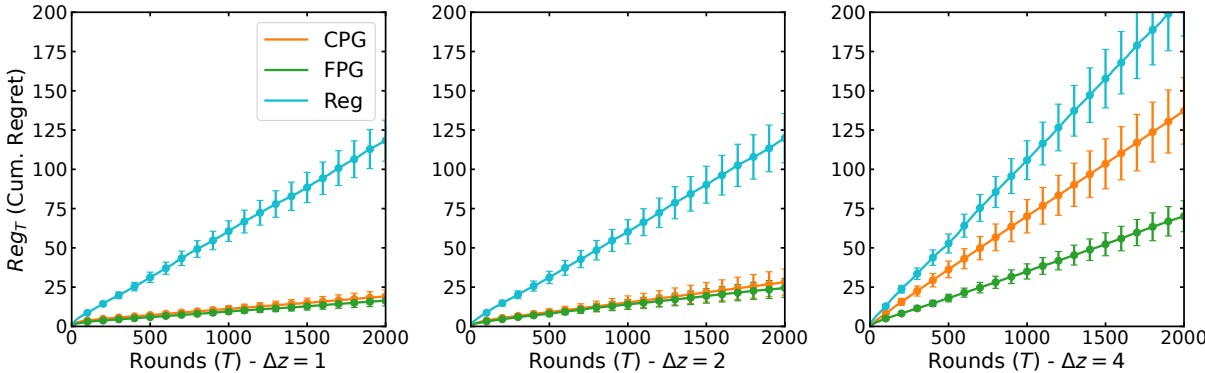

Figure 9: Expected cumulative regret for bandit algorithms for out of distribution generalization with means $\Delta z = 1, \Delta z = 2, \Delta z = 4$. Error bars indicate one standard error across 200 seeds.

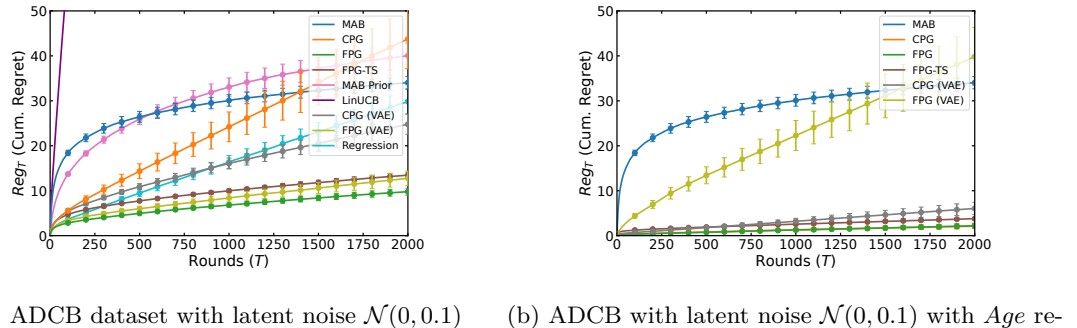

(a) ADCB dataset with latent noise $\mathcal{N}(0, 0.1)$

(b) ADCB with latent noise $\mathcal{N}(0, 0.1)$ with *Age* related variable removed from the dataset.

Figure 10: Expected cumulative regret for bandit algorithms for the respective ADCB data with latent noise $\mathcal{N}(0, 0.1)$. Error bars indicate one standard error across 200 seeds.

### E.3 Noise in the latent distribution and ADCB performance

To study the effect of latent noise, we increased the latent noise variance in the ADCB dataset from $\mathcal{N}(0, 0.02)$ in the main paper to $\mathcal{N}(0, 0.1)$. We then refit the LVM-based models and report the resulting bandit performance in Figure 10a. The increased latent noise makes recovery of the latent state more difficult, placing the LVM-based methods, especially `CPG`, at a disadvantage. As the reward noise remains unchanged, methods that ignore contextual information, such as the MAB baseline, are unaffected.

We see in Figure 10a that VAE model performs better compared to `ILB`-based models in online, and in test performance in Table 5. We notice that this may be due to the *Age* variable in the ADCB dataset which has unique value for each patient making it trivial to predict the patient index ($C$ in 4). After removing the *Age* variable from the dataset and refitting the LVMs, we measured the performance. We noticed a considerable increase in test results and bandit performance; and the results are in Figure 10b.

### E.4 Noise in the reward

The results in Figure 11 show the performance of LVM based `CPG`, `FPG` and `FPG-TS` models compared to the regression model and a Thompson sampling based MAB under increasing reward variance. The MAB baseline relies solely on observed rewards and takes longer to converge when the variance increases. In comparison, `FPG` model is robust and converges slowly to the `CPG` model in performance.

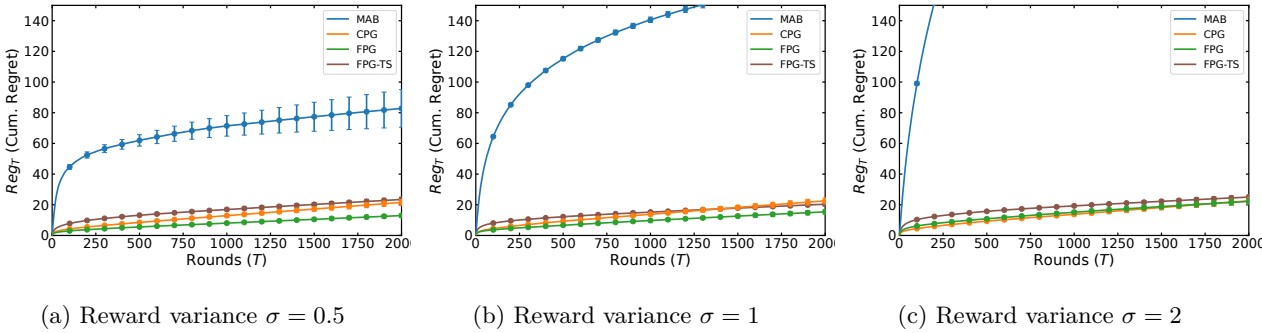

(a) Reward variance $\sigma = 0.5$     (b) Reward variance $\sigma = 1$     (c) Reward variance $\sigma = 2$

Figure 11: Expected cumulative regret for bandit algorithms on Synthetic dataset with increasing noise in the reward. Error bars indicate one standard error across 200 seeds.

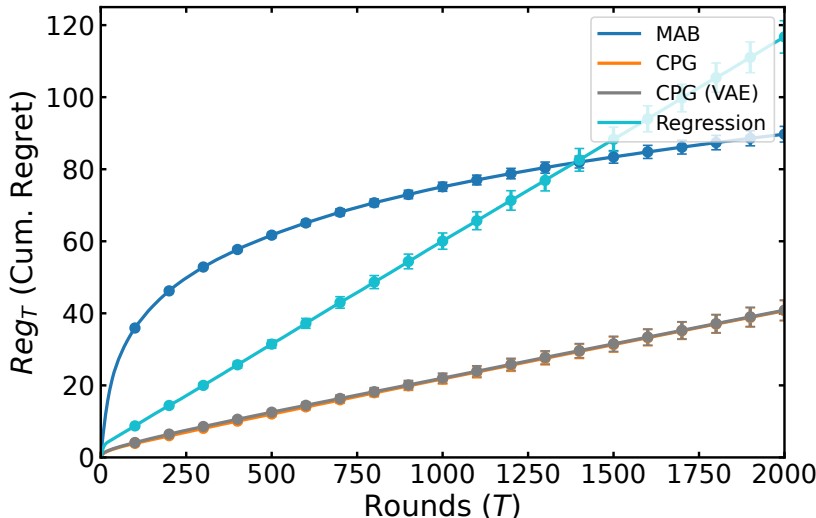

Figure 12: Expected cumulative regret for MAB, Regression and `CPG` algorithms for the nonlinear reward function. The error bars indicate one standard error computed across 1000 seeds.

### E.5 Nonlinear reward model

In Assumption 3.1 and Theorem 3.4 we assume linearity for the function that $\theta$ determines the rewards from the latent state $Z_i$. This assumption is indeed useful as it allows for exploration by giving a close-form result for the posterior for `FPG`. However, our `ILB` algorithm also works when we allow for $\theta$ to be nonlinear. In Figure 12 we used a randomly initialized two layered MLP with leaky ReLU activations for the reward function $\theta$. During the offline phase, we fit a four layered MLP to the estimated $Z_i$. For the fitted MLP we used leaky ReLU activations except for the final layer, which had no activations. For training we used Adam optimizer with learning rate 0.001 and weight decay 0.0001, and trained until convergence. For the offline stage, we only used the greedy strategy with `CPG`. The results show the effectiveness of our approach compared to the regression baseline. VAE based `CPG` performs similarly to our approach when compared to the linear case Figure 4.

### E.6 Generalization to exponential family

In Figure 13, we conduct experiments where the stationary noise in the latent state is distributed with respect to Laplace and uniform distributions in order to show model performance under exponential family noise.

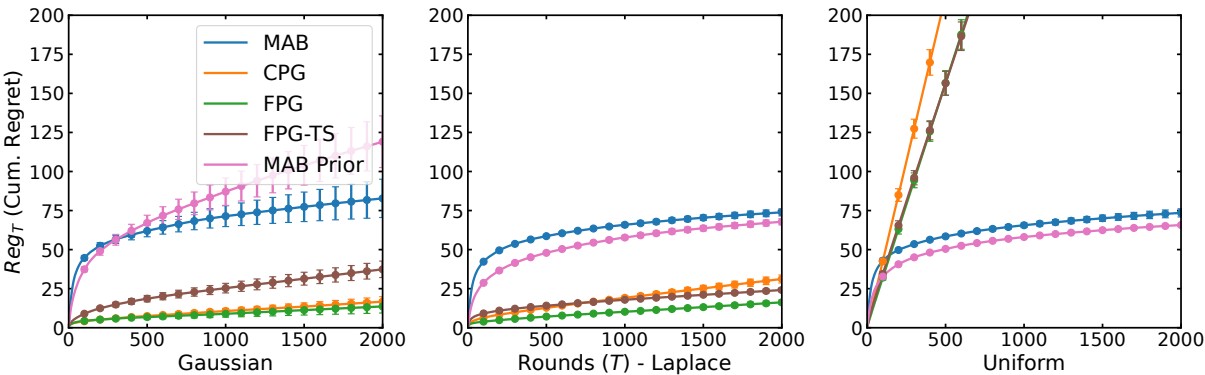

Figure 13: Expected cumulative regret for different exponential family noise. The error bars indicate one standard error computed across 1000 seeds.

LVM based models perform poorly due to the high variance of the uniform distribution, but outperform the Gaussian case for the tightly concentrated Laplace distribution. The results are comparable to having different levels of noise in the latent state.

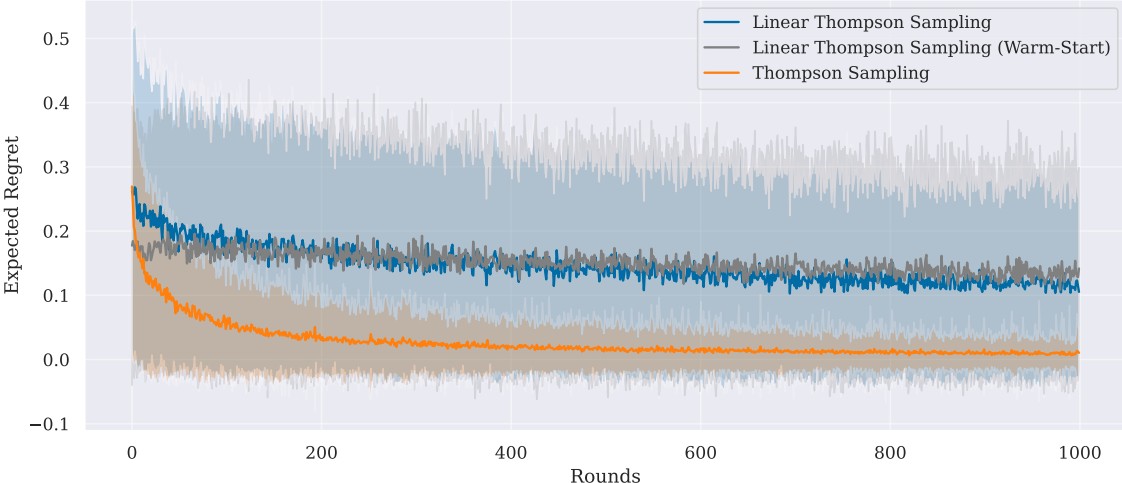

Figure 14: Synthetic example comparing linear contextual bandits for stationary context. $K = 10, d = 5, 500$ warm-start samples

## F  Contextual Bandits in Our Setting

As discussed in Section 2, in our setting the optimal action $a_i^*$ depends on the context $X_{i,t}$ only through the latent variable $Z_i$. Since $Z_i$ is fixed over time, the optimal action $a_i^*$ remains constant even as the context $X_{i,t}$ changes. Consequently, a contextual model that relies solely on $X_{i,t}$ cannot exploit the reward structure determined by $Z_i$. If the learner lacks an estimate of the latent variable, the context $X_{i,t}$ does not provide additional information beyond the observed reward $R_{i,t}$. The learning problem then reduces to estimating each reward distribution $\mu_i$ from the observed rewards $R_{i,t}$, and choosing the action with the highest expected reward:

$$a_i^* = \arg \max_{a \in \mathcal{A}} \ \mathbb{E}[R_{i,t}|A = a],$$

which is a non-contextual multi-armed bandit problem.

An extreme example of this is when the observed context $X_{i,t}$ is stationary for all $t \in [T]$ with a fixed $\theta_A \in \mathbb{R}^{|\mathcal{A}| \times d}$, for $d$-dimensional context. We illustrate this empirically with a synthetic example:

Synthetic arm parameters:

$$\theta_{a,d} \sim \mathcal{U}(0.3, 0.8), \quad \epsilon_{a,d} \sim \mathcal{N}(0, 0.25), \quad \forall a \in \mathcal{A}$$

Synthetic context:

$$X_i \sim \mathcal{N}(\mu, \Sigma), \quad \forall i \in \{1, 2, \ldots, N\}.$$

$$\mu = \begin{bmatrix} 0.5 \\ 0.5 \\ \vdots \\ 0.5 \end{bmatrix} \in \mathbb{R}^d, \quad \Sigma = 0.1\mathbf{I}_d + 0.05 \cdot \text{triu}(\mathbf{1}_{d \times d}, 1) + 0.05 \cdot \text{tril}(\mathbf{1}_{d \times d}, -1),$$

where:

- $0.1\mathbf{I}_d$ : Diagonal matrix with variance 0.1.

- $0.05 \cdot \text{triu}(\mathbf{1}_{d \times d}, 1)$ : Upper triangular part (excluding diagonal) filled with 0.05.

- $0.05 \cdot \text{tril}(\mathbf{1}_{d \times d}, -1)$ : Lower triangular part (excluding diagonal) filled with 0.05.

As seen in Figure 14, a non-contextual Thompson sampling (Thompson, 1933; Russo et al., 2018) algorithm outperforms its contextual (Agrawal & Goyal, 2013) counterpart. We also include a warm-started Thompson sampling algorithm (Oetomo et al., 2023) which suffers the same fate. Warm-started Thompson sampling however converges faster than the not warm-started contextual Thompson sampling, although the non-optimal convergence for both is fairly similar compared to the non-contextual Thompson sampling.

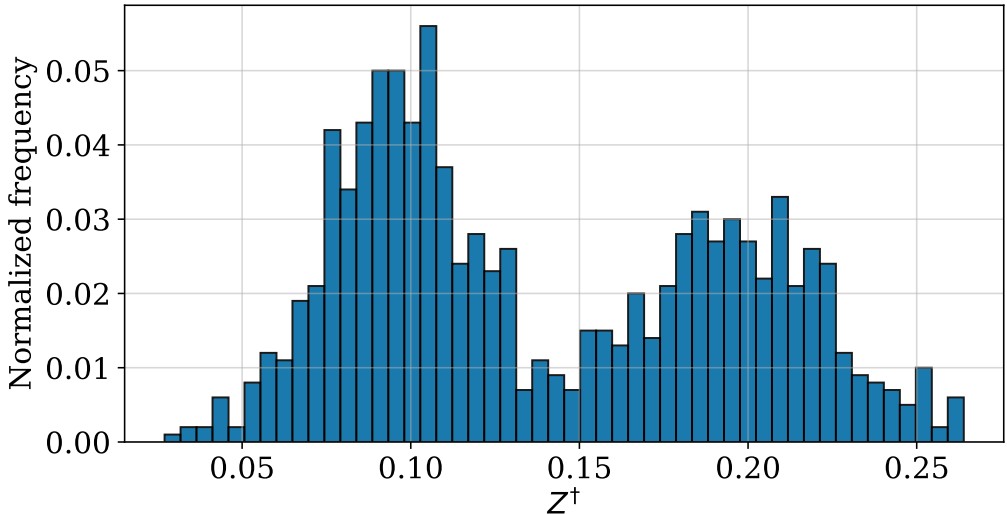

Figure 15: Histogram over 50 bins of the bimodally distributed continuous component of the latent state in ADCB

## G Conditional Reward modeling in ADCB from Average Treatment Effects

We aim to model a conditional treatment effect function $\text{ATE}_a(z)$ for each treatment $a$ such that the expected treatment effect over the distribution of a continuous latent state $Z = Z^\dagger$ (where $Z^\dagger$ is the continuous component of the latent state in ADCB) matches a predefined set of average treatment effects (ATEs). We'd also like to have heterogeneity of the treatments over $Z$. We model this on a latent state whose distribution is bimodal as shown in Figure 15.

### G.1 Gaussian Mixture Model

Given our continuous latent state $Z$, its distribution can be expressed as a Gaussian Mixture Model (GMM) with two components:

$$p(Z) = \lambda_1 \mathcal{N}(\mu_1, \sigma_1^2) + \lambda_2 \mathcal{N}(\mu_2, \sigma_2^2)$$

where:

- $\lambda_1 = 0.572$ and $\lambda_2 = 0.428$ are the mixture weights with $\lambda_1 + \lambda_2 = 1$,

- $\mu_1 = 0.0979$ and $\mu_2 = 0.1986$ are the means of the Gaussian components,

- $\sigma_1^2 = 0.000541$ and $\sigma_2^2 = 0.000752$ are the variances of the Gaussian components.

### G.2 Expected Value of Z

The expected value of the latent state $Z$ under this bimodal distribution is given by:

$$\mathbb{E}[Z] = \lambda_1 \mu_1 + \lambda_2 \mu_2$$

Substituting the values, we find:

$$\mathbb{E}[Z] = (0.572)(0.0979) + (0.428)(0.1986) \approx 0.1403$$

### G.3 Heterogeneous Treatment Effect Model

The treatment effect for each treatment $a$ is assumed to be a linear function of $Z$:

$$\text{ATE}_a(Z) = \alpha_a Z + \gamma_a$$

where $\alpha_a$ and $\gamma_a$ are treatment heterogeneity parameters to be determined. The values of $\gamma_a$ are chosen and fixed as:

$$\gamma = [0, -0.5, -1, -0.5, -2, -3.5, -1, -2.9]$$

### G.4 Expected Treatment Effect

The expected treatment effect for each treatment $a$ over the distribution of $Z$ is given by:

$$\mathbb{E}_Z[\text{ATE}_a(Z)] = \mathbb{E}[\alpha_a Z + \gamma_a] = \alpha_a \mathbb{E}[Z] + \gamma_a$$

### G.5 Matching Expected Treatment Effects

We want the expected treatment effect for each treatment $a$ to match a predefined average treatment effect $A_\Delta(a)$. We use 8 treatments with given values for $A_\Delta$:

$$A_\Delta = [0, 1.95, 2.48, 3.03, 3.20, 2.01, 1.29, 2.69]$$

This gives:

$$\alpha_a \mathbb{E}[Z] + \gamma_a = A_\Delta(a)$$

We can solve for $\alpha_a$ as:

$$\alpha_a = \frac{A_\Delta(a) - \gamma_a}{\mathbb{E}[Z]}$$

### G.6 Matching Expected Treatment Effects with Noisy ATEs

To account for noise in the treatment effect observations, we introduce a level-variable additive Gaussian noise $\zeta_a \sim \mathcal{N}(0, \sigma^2)$:

$$\alpha_a \mathbb{E}[Z] + \gamma_a = A_\Delta(a) + \zeta_a$$

Solving for $\alpha_a$, we find:

$$\alpha_a = \frac{A_\Delta(a) + \zeta_a - \gamma_a}{\mathbb{E}[Z]}$$

For a given value of $z$, the conditional treatment effect is then computable as:

$$\text{ATE}_a(z) = \alpha_a z + \gamma_a - \zeta_a$$

Using $\text{ATE}_a(z), \forall a \in \mathcal{A}$ as a conditional reward model gives us a model of $\mathbb{E}[R \mid Z = z, A = a] = \text{ATE}_a(z)$. The resulting conditional reward models are illustrated in Figure 16.

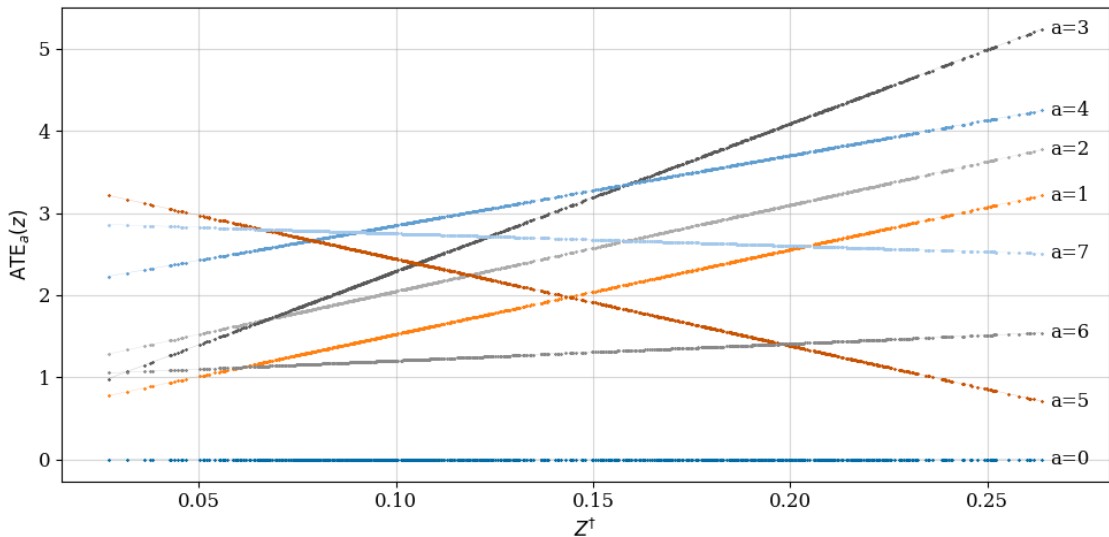

Figure 16: Conditional linear reward models in ADCB with heterogeneity over the latent state.

## H   Experimental Setup

**Training details for LVM**   We use an MLP with maxout activation functions for the feature extractor $f$. We select $L = 2$ and $L = 4$ layered models equal to different settings described in the data generating process, with hidden dimensions equal to dimensionality of $Z_i$. For the LVM we do a two-stage training: First, we freeze the MLP weights and only train the linear classifier, and then we train MLP and the classifier together. We train the MLP using SGD with momentum and $\ell_2$-regularization with initial learning rate of 0.01, exponential decay of 0.1, and momentum 0.9. We run each experiment across 10 different seeds.

**Training details for VAE**   We select $L = 2$ and $L = 4$ layered encoder and decoder models equal to different settings described in the data generating process, with hidden dimensions equal to dimensionality of $Z_i$. We train the model with KL-divergence reconstruction loss. For training we use Adam optimizer with a weight decay of 0.001 and a learning rate of 0.001. We train for 100 epochs with early stopping based on validation loss with a patience of 5 epochs. We run each experiment across 10 different seeds.

**Details for the regression baseline**   For the regression baseline, we use a TARNet architecture (Shalit et al., 2017) with a GRU encoder using $L = 2$ and $L = 4$ layers for different settings with a hidden feature size of 64, selected on the validation set. We train the model with MSE loss on observed rewards using Adam optimizer with weight decay of 0.001 and an exponentially dampening learning rate starting at 0.01 with decay factor 0.1. We train for 100 epochs, each with increasing sequence length and a batch size of 100 instance sequence and perform early stopping based on $R^2$ for observed rewards. We run each experiment across 10 different seeds.

**Further training details**   We used an NVIDIA T4 GPU for producing most of the training for this work. Most expensive experiments took at most 2 hours train for 10 seeds. For bandit algorithms we used 10 Intel(R) Xeon(R) Gold 6338 CPUs and ran seeds in parallel with a cost of about 5 CPU hours for 200 seeds.

**Computational complexity of Algorithm 1**   At round $t$ our algorithm uses a forward pass through the model $f$ in Line 3. The time complexity of a forward pass depends on the dimension of $x_{i,t}$, $n$; number of layers, $L$, and the number of neurons in each layer, $h_l$. We then add to a running average of estimates, which has complexity $O(1)$. For `CPG` we then select greedy actions with arg max, which has time complexity relative to number of actions $O(|\mathcal{A}|)$. For FPG we use a LBFGS optimizer to solve Equation (7) of Algorithm 1,

which has a time complexity that depends on number of dimension $n$ and number of iterations. Finally for FPG, we either select greedy actions $O(|A|)$, or sample using Line 9.

## I   LVM Results

Complete results on the test set for fitting of LVM, VAE, and Regression baselines for ADCB and LVM datasets Table 5.

Table 5: Complete LVM fitting results. $L$ layers in the MLP, $T_o$ time steps.

| $L$ | $T_o$ | Model | $\mathrm{MCC}_Z$ | $R^2_R$ | % $a^*$ |
|---|---|---|---|---|---|
| **Synthetic Data** | | | | | |
| 2 | 100 | LVM | 0.89 | 0.75 | 79 |
| 2 | 200 | LVM | 0.92 | 0.76 | 82 |
| 2 | 300 | LVM | 0.91 | 0.75 | 87 |
| 4 | 100 | LVM | 0.91 | 0.72 | 76 |
| 4 | 200 | LVM | 0.90 | 0.75 | 82 |
| 4 | 300 | LVM | 0.91 | 0.74 | 82 |
| 2 | 100 | VAE | 0.94 | 0.69 | 70 |
| 2 | 200 | VAE | 0.93 | 0.72 | 88 |
| 2 | 300 | VAE | 0.94 | 0.70 | 72 |
| 4 | 100 | VAE | 0.87 | 0.43 | 40 |
| 4 | 200 | VAE | 0.91 | 0.43 | 48 |
| 4 | 300 | VAE | 0.91 | 0.51 | 52 |
| 2 | 100 | Regression | - | 0.70 | 61 |
| 2 | 200 | Regression | - | 0.75 | 69 |
| 2 | 300 | Regression | - | 0.72 | 73 |
| 4 | 100 | Regression | - | 0.50 | 29 |
| 4 | 200 | Regression | - | 0.61 | 44 |
| 4 | 300 | Regression | - | 0.59 | 55 |
| **ADCB** | | | | | |
| 2 | 100 | LVM | - | 0.92 | 88 |
| 2 | 200 | LVM | - | 0.92 | 89 |
| 2 | 300 | LVM | - | 0.92 | 87 |
| 4 | 100 | LVM | - | 0.91 | 86 |
| 4 | 200 | LVM | - | 0.90 | 78 |
| 4 | 300 | LVM | - | 0.91 | 82 |
| 2 | 100 | VAE | - | 0.94 | 95 |
| 2 | 200 | VAE | - | 0.94 | 97 |
| 2 | 300 | VAE | - | 0.94 | 95 |
| 4 | 100 | VAE | - | 0.94 | 95 |
| 4 | 200 | VAE | - | 0.94 | 96 |
| 4 | 300 | VAE | - | 0.94 | 95 |
| 2 | 100 | Regression | - | 0.95 | 89 |
| 2 | 200 | Regression | - | 0.95 | 92 |
| 2 | 300 | Regression | - | 0.95 | 93 |
| 4 | 100 | Regression | - | 0.89 | 51 |
| 4 | 200 | Regression | - | 0.95 | 90 |
| 4 | 300 | Regression | - | 0.95 | 93 |

