# OpenReview forum: "Identifiable Latent Bandits: Leveraging observational data for personalized decision-making"
_TMLR — Accepted by TMLR_

### Review · Reviewer_BJvW · 2026-02-22

**Summary Of Contributions:**

This paper addresses sample inefficiency in personalized sequential decision-making by learning latent representations from offline observational data to improve contextual knowledge during online learning. The core premise is that instances share a latent state that (i) is constant over time for the instance and (ii) determines action rewards, while observed contexts are generated from noisy latent observations via an injective nonlinear emission function. It introduces identifiable latent bandits, ILB, the first family of latent bandit algorithms which learns latent state without a known Latent Variable Model (LVM) assumption. The LVM is learned (up to an invertible affine transform based on Theorem 3.3) via the mean contrastive learning method introduced during offline stage. It is argued theoretically and empirically that the identifiability of the posterior of latent variable and interventional reward distributions are critical to guarantee the success of ILB algorithms without potential compounding or biases.

Strength:
- Tackles a real and important bottleneck: per-instance horizons are small, so purely online bandits can be infeasible in settings like medicine.
- Clear conceptual advance over prior latent bandit work that assumes the LVM is known: it formalizes how to learn an identifiable latent representation from observational logs.
- Despite stringent assumptions, this work produces theoretical guarantees that are mathematically rigor, which offers confidence for researchers developing new algorithms in this direction.
- Good clarity on all the theorems and motivations. Proper justifications are provided on choices and assumptions made, and the reasonings are convincing. The pipeline and the algorithms are easy to follow via clean diagrams and well-annotated pseudo-code.

Weakness:
- The strongest guarantees rely on stringent assumptions: stationary latent state, injective g, known latent dimension n≤d, product distribution assumptions, and (for causal reward identification) linear rewards.
- The regret guarantee (Theorem 3.5) assumes an “optimal model pair” (essentially perfect recovery up to affine transform), which is not realistic; there is no finite-sample bound linking offline estimation error to online regret.
- The offline step requires many observations per instance (the identifiability theorem is in the per-instance infinite data limit and the ambiguity of LVM learning diminishes with sufficiently large data sets), which may conflict with the motivating regimes (few decisions per patient) and requires justification.
- The ADCB experiments note an “Age” feature issue (apparently near-unique per instance) that can make instance classification trivial and may inflate representational recovery; this deserves clearer treatment in the main narrative.

**Audience:**

Yes

**Audience Explanation:**

It connects identifiable representation learning (nonlinear ICA / contrastive learning) with bandits/personalized decision-making, a combination that is likely to interest both theory-leaning and methods-focused ML readers.
The offline+online personalization setting is practically relevant (health, recommender systems, etc), and the paper provides a concrete algorithmic pipeline plus sensitivity experiments to identify boundaries and limitations, which may motivate further investigations in this direction.

**Broader Impact Concerns:**

The paper includes a brief Broader Impact Statement emphasizing methodological intent. Given that the data and experiments are run mainly in a synthetic or semi-synthetic setting, it raises little or no ethical concern.

**Claims And Evidence:**

Yes

**Claims Explanation:**

The paper provides a coherent set of identifying assumptions (Assumption 3.1–3.2), then proves representation identifiability up to affine transform (Theorem 3.3).
The key issues of identifiability of the posterior of latent variable and interventional reward distributions are address under the assumptions and the linear-case identifiability conclusion (Theorem 3.4).
Empirical results (Table 1, Figures 3–7) show that when assumptions hold, learned ILB models enable faster convergence than online-only baselines and can outperform non-identifiable representation baselines in several regimes.

**Requested Changes:**

- Several headline claims (e.g., “shorter exploration time than classical bandits” broadly) are only guaranteed under strong correctness assumptions (perfect or near-perfect LVM). The current theory does not quantify how errors in $f$ and $\hat{\theta}$ degrade online performance.
- Theorem 3.3 is stated in the limit $T_i \to \infty$ the paper should either (i) justify why moderate $T_i$ suffices in the target applications in practice, or (ii) reposition the contribution as primarily foundational, with explicit discussion of practical sample size needs.
- The paper notes that Age can make predicting patient identity trivial and affects ILB performance; this is important because the offline learning objective is literally instance classification. This should be highlighted where ADCB is introduced, with the cleaned setting reported as primary or at least co-primary.

---

> ### Author Response · Authors · 2026-03-13
> **Responses by Authors**
>
> _We thank the reviewer for their insightful and constructive comments on our paper. We uploaded a revised version of our paper with additions highlighted in blue. We also respond to a few points below:_
>
> ---
> ### 1.  Re: Claims of sample efficiency
>
> - We believe you are referring to the claims made in the contributions and the abstract. These claims are not based solely on our theoretical results, but are also supported by our experimental findings. In particular, our CPG and FPG algorithms consistently exhibit lower regret compared to purely online MAB baselines and offline-online MAB with learned prior (MAB Prior), both when the assumptions of our analysis hold and when they are violated.
> - Regarding the assumptions in Theorem 3.5 (“perfect or near-perfect LVM”) and the lack of analysis linking offline estimation error to online regret, we agree that such an end-to-end finite-sample analysis would be interesting. However, proving such a result is non-trivial, as errors in the learned representation can affect instances differently. As a result, a bound on the expected contrastive loss from finite samples is generally insufficient to bound the error in recovering a fixed latent state; establishing a regret bound would instead require a uniform error bound. We discuss this point in more detail in Section 2 (Re: Thm 3.5 Optimal Pair).
>
> **Changes:**
> - We add a revised version of Theorem 3.5 incorporating errors in the learned model. (see 2. Re: Thm 3.5 Optimal Pair)
> ---
> ### 2. Re: Thm 3.5 Optimal Pair
>
> In Latent Bandits Revisited [1], the authors prove in Theorem 2 that if deviations in reward estimates are uniformly bounded by a constant $\epsilon$, this adds linear regret to their bound with other terms maintained. We  prove an analogous result for CPG, assuming that for a learned model pair $f, \theta, \forall a, z:\left|\hat{\theta}^{\top} f(g(z))-\theta^{\top} z\right|<\epsilon$ leads to the regret bound for CPG, presented in our revision.
>
> We are unaware of uniform guarantees for contrastive learning that would complement this result, but welcome suggestions! In [1], the authors appeal to spectral methods for such bounds, but the cited works are limited to linear LVMs. For instance in [2], the authors give a bound for linear ICA with white noise : $x=A z+\eta$, and show a guarantee of type $P(\|A-\hat{A}\|<\epsilon)>1-\delta$ using their proposed estimator. Although, such results can enable an end-to-end analysis, we stress that it does not generalize to nonlinear functions.
>
> ---
> ### 3. Re: Implications of Theorem 3.3
> Theorem 3.3 concerns the identifiability of the inverse emission function $g^{-1}$ and the result is not directly connected to the sample efficiency of a learning algorithm. We can present the optimal feature extractor $f^*$ as maximizing the log-likelihood objective
>
> $$f*, q* = \arg\max_{f, q} \sum_{i =1}^I \log p(C=i|X=x; f,q)p(C=i).$$
>
> A universal function approximator, will maximize this objective in the infinite-sample limit. In practical applications, the sample complexity of the dataset $\mathcal{D}$ required to learn $f$ is difficult to characterize. It depends on the dimensions of $z$, and excat the form of the emission
>
> In real world applications, the sample complexity of the dataset $\mathcal{D}$ needed to learn $f$ is not trivial to characterize, as it depends on the number of dimensions, $n$ of $z$ and the exact form of the emission function $g$. It is necessary to have $I \geq n$ instances as stated in Assumption 2 (b). In our experiments we observed that $100$ per-instance observations are sufficient for learning, when the emission function $g$ is a $4$ layered MLP (see Table 4). However, our experiments are insufficient to characterize sample needs as different data modalities have different needs.
>
> **Changes:**
> - Changed the presentation of max-likelihood objective
> ---
> ### Re: 4. About ADCB Data
> - Yes, this is an important point. We intentionally highlight the *Age* variable in the ADCB to illustrate a limitation of our instance classification method. When training instances differ by near-trivial attributes, contrastive learning algorithms can struggle to learn meaningful representations. We note that the ADCB dataset does not satisfy our assumptions, namely the structural equations and the noise in Assumption 3.1 (a) is violated.
> - We  revise the paragraph where we introduce ADCB, and moved the discussion about the *Age* variable.
>
> **Changes:**
> - We made changes to the paragraph where we introduce ADCB to reflect how we treat the $Age$ variable in the experiments.
> ---
> [1] Hong et al., Latent Bandits revisited, NeurIPS 2020.
>
> [2] Anandkumar, Animashree, et al. "Tensor decompositions for learning latent variable models." _J. Mach. Learn. Res._ 15.1 (2014): 2773-2832.
>
> ---
> _We hope that our response addresses your concerns and are happy to engage in further discussion! --Authors_

---

### Review · Reviewer_vy5c · 2026-03-04

**Summary Of Contributions:**

The paper proposes a methodology to learn latent representations from observational data and leverage them for personalized decision-making in latent bandit settings.
The authors defines identifiable latent bandits which aims to solve the problem of latent bandits with continuous latent state without knowing latent variable model as a prior.
The author proposes a two-stage method to learn the latent representation in offline dataset and then use it for online decision making. The paper provides theoretical guarantees for the proposed method and demonstrates its effectiveness through experiments on synthetic datasets.

**Additional Comments:**

Please refer to the requested changes.

**Audience:**

Yes

**Audience Explanation:**

Yes, I believe that individuals in TMLR's audience would be interested in knowing the findings of this paper. The problem of latent bandits with continuous latent state is a challenging one, and the proposed methodology offers a novel approach to address this problem. The theoretical guarantees provided in the paper, as well as the experimental results, demonstrate the effectiveness of the proposed method, which could be of interest to researchers and practitioners in the field of machine learning, particularly those working on bandit problems and personalized decision-making.

**Broader Impact Concerns:**

There is no concern on the ethical implications since this is a theoretical work with synthetic experiments.

**Claims And Evidence:**

Yes

**Claims Explanation:**

The authors propose a novel approach to address the problem of latent bandits with continuous latent state without knowing latent variable model as a prior. The paper provides theoretical guarantees for the proposed method and demonstrates its effectiveness through experiments on synthetic datasets. However, the paper could benefit from clearer explanations in some sections, particularly in Section 3.2 where the methodology is described. Additionally, it would be helpful if the authors could clarify the motivation behind their hybrid approach and how it addresses the problem of online learning in this context.

**Requested Changes:**

I would like to see the following changes in the paper:

1. Section 3.2 is a bit hard to follow. It would be helpful if the authors could use notations that are more consistent with the rest of the paper and provide more intuition about the methodology applied in this section. For example, a high level overview of ICA and how it is applied in this context would be helpful for readers who are not familiar with this technique.

2. The author proposes an algorithm including two stages, offline learning to learn the latent representation and online decision making. It would be helpful if the authors could state this hybrid approach as a problem setting or only as the solution to an online learning problem. This would help to clarify the motivation behind the proposed method and make it easier for readers to understand the problem being addressed.

3. In the algorithm statement, the authors list four if-conditions to handle different setting. It would be helpful if the authors could clarify the motivation behind these conditions and how they relate to the problem of online learning in this context. For example, it would be helpful to explain when CPG is chosen and when FPG is chosen, and how these choices affect the performance of the algorithm.

4. In the algorithm statement, I find that there is a Greedy policy is used for online decision making. It is commonly known that Greedy policy is not a good choice for online learning. It would be helpful if the authors could clarify why the Greedy policy works in this context. For example, is it because the offline learning stage provides a good latent representation that allows the Greedy policy to perform well? Or is it because the problem setting is such that the Greedy policy is sufficient for achieving good performance, like identifiability?

---

> ### Author Response · Authors · 2026-03-13
> **Responses by Authors**
>
> _We thank the reviewer for their insightful and constructive comments on our paper. We upload a revised version of our paper with additions highlighted in blue. We also respond to a few points below:_
>
> ### 1. Re: Clarifications to Section 3.2
> We made changes to Section 3.2  to add clarifications throughout the section. We also added a high-level overview of identifiable representations in the first paragraph. We switched the paragraph's focus from ICA to identifiable representations as it has a broader focus and we already introduce the literature in the introduction. We hope it creates a more cohesive text and that you are more satisfied with the current version.
>
> **Changes:** Section 3.2
>
> ---
> ### 2. Re: Hybrid Approach
> Just to clarify, we use the term _hybrid algorithms_ to refer to algorithms that combine offline and online learning for adaptive decision-making. We do not present the hybrid approach as a novel problem setting; rather, it arises naturally in latent bandits due to the need to learn a latent variable model (LVM) from observational data. There are previous works using a similar paradigm, which we mention in our literature review, namely latent bandits (where a LVM is assumed to be learned from data), warm-starting model parameters, and incorporating learned priors.  In our experiments, we compare against offline (regression), online (MAB), and offline and online combined methods (MAB with a learned prior). We would be happy to revise the wording to make this more clear if you could point us to any specific statements that may be ambiguous.
>
> ---
> ### 3. Re: Comparison of CPG vs FPG
> - Thanks for bringing this point to our attention. We added a new paragraph discussing the benefits of FPG over CPG in the case of model misspecification to Section 3.3 (after Theorem 3.5) with a recommendation for FPG. It now reads as follows:
>
> In the case that the LVM is misspecified or misestimated, greedy reward maximization with respect to $\hat{z}$ as in Algorithm 1 will be biased in general and yield linear cumulative regret. In this case, one could either try to re-estimate the arm parameters $\theta$, or update the latent variable estimate $\hat{z}_i$ at inference time. These choices are equivalent as the reward is bilinear. An example of the latter is to look at the reward history and search for the latent $\hat{z}_i$ which best explains the previous rewards and contexts, conditioned on arm parameters. This trades off exploiting the inference function $f$ and explaining previous rewards. Our second algorithm, _full posterior greedy_ (FPG), takes this approach and minimizes the full negative log-likelihood (8) to choose the best action. We recommend FPG as a more practical and adaptive alternative to CPG. In Lemma D.1, we analyze the mean and variance for FPG estimates of $z_i$.
>
> - Also added small changes to discuss the benefits of exploration to the paragraph where we discuss FPG-TS. (last paragraph of 3.3, now called **Exploration**)
>
> Once a latent is estimated, we could either use a greedy strategy and choose the best arm under the estimated latent state, or use the posterior for the estimated reward $\hat{\mu}_i = \hat{\theta}^\top \hat{z}_i$ for exploration. For FPG this can lead to choosing arms that reduce the variance of the estimate (8) and recover from a biased LVM estimate.
>
> We hope these changes address your concerns.
>
> **Changes:** Section 3.3
>
> ---
> ### 4. Re: Greedy policy
> Yes, you are absolutely right that a greedy policy is generally not a good choice for online learning. However, under our assumptions and due to our identifiability result, the greedy CPG algorithm turns out to be a reasonable strategy and suffers only constant regret. This would not be the case without an identifiable LVM.  We also highlight this in the paragraph added in response to Point 3 (Re: Comparison of CPG and FPG), where we note that the greedy CPG algorithm may suffer linear regret in the worst case when the LVM is misspecified. In practice, we recommend FPG as a more practical and adaptive alternative to CPG.
>
> **Changes:** Section 3.3
>
> ---
> _We hope that our response addresses your concerns and are happy to engage in further discussion! --Authors_

---

### Review · Reviewer_eTaT · 2026-03-04

**Summary Of Contributions:**

This paper proposes the Identifiable Latent Bandits (ILB), where a continuous vector-valued latent state is unknown and only the context $ X_t$ is observed.   Here The $X_t=g(Z_t)$ is generated by a hidden mechanism.
The paper considers the utilization of the offline data: data sets from several patients, which are used to train the model to infer the relationship of latent state $Z \in \mathbb{R}^n$ and context $X$. In the online part (for a fixed user $i$), at each round, the learner observes the context $X_t$ , and then infers $Z_t$ by the trained model $f^*$ from offline data. Then she will take an action $A_t$ from $K$ actions and gain the (linear) reward.


From the offline data, they propose the Mean-constrative learning based on non-linear ICA theory to learn the $g^{-1}$. The classifier $f^*$, to predict to which
instance $c \in  [I]$ an observation belongs, learned from offline observational data $\mathcal{D}$, will convergel to the inverse emission
function$g^{-1}$ as the number of observations per instance approaches infinity up to an invertible affine transformation.
Based on this, the action selection is done via a greedy policy. Under some assumptions, the simple greedy algorithm obtains a constant regret independent of $T$. The experiments are based on synthetic and ADCB (Kinyanjui & Johansson, 2022). CPG and its variant FPG outperform over baselines.

**Additional Comments:**

N/A

**Audience:**

Yes

**Audience Explanation:**

- The use of constrative-learning in Latent Bandits to deal with unknown latent structure is an interesting and novel proposal.

- Under some assumptions, the simple greedy algorithm obtains the constant regret independent of $T$.

**Claims And Evidence:**

No

**Claims Explanation:**

There are weaknesses in the paper:

- The optimality guarantee of $f^*$ in Theorem 3.3 is based on a very strong assumption: infinite per-instance data. If this is difficult to address, it is the major weakness of this paper. Under a finite per-instance data setting (where the error rate depends on $\sum_{i \in I}T_i$), constant regret is likely unachievable, making the online part much harder. In other words, assuming infinite offline data drastically simplifies the online problem. In both practice and standard bandit problems, it is important to know how much offline data is required to make online learning stable. Strengthening this theoretical guarantee is essential, as iIdentifiability of the posterior of $Z$ is easily achievable as we have infinite offline data $\mathcal{D}$, which was the central challenge of the paper.

- The authors mention Hong et al. (2020)  and stated that “it assume that both models are known a priori but give little guidance for how to learn or acquire them.” However, both mmUCB and mmTS are designed for model misspecification and they also provide regret bounds.

- Offline data $\mathcal{D}$ of $I$ previous problem instances requires the matrix of patient latent states to be $n$-rank. In high dimentional case where $n$ is large, this assumption might be strong. Since latent states are hidden to the learner, it is not easy to verify if the assumption is true or not in advance.

- The theoretical analysis is only applicable to the linear reward $\theta_a^{\top}z$.


-  In Theorem 3.5., we need to assume that $\| \theta_a\|_2=\|\hat{\theta}_a\|_2=1$ for all $a$. This is another strong assumption which is hard to verify.

**Requested Changes:**

Questions

- Why is $B^{-1}$  not included in the final regret bound?
The noise term $\eta_t^{‘}$ is multiplied by $B^{-1}$, and it would affect the norm of variance.

- I cannot fully read the assumption of Theorem 3.5. $\hat{\theta}$ is obtained via inferred $\hat{z}$ which is scaled by a $B$ due to the affine transform difference associated with $f$. So we have $\hat{\theta}_a = B^\top \theta_a$. However, the proof of Theorem 3.5 assumes that $||\theta_a||_2  =||\hat{\theta}_a||_2 = 1$ for all actions. $B$ should be strictly orthogonal to satisfy this condition.  Could you explain around this assumption?


Request Changes
- Add precise discussions; how it is different from Hong et al. (2020) in the problem setting, assumptions, and regret bounds.

 - I recommend to include Remark on Theorem 3.4  in the main text. The fact that the optimal actions or ranking is identifiable is more intuitive to explain why the second online part is successful.

- See Weakness and Questions for other points.c


Typo:
$a_t$ in Section 3.3 will be defined as argmax $ \theta_a^{\top} \hat{z}_{i,t}$

---

> ### Author Response · Authors · 2026-03-13
> **Responses by Authors**
>
> _We thank the reviewer for their insightful and constructive comments and taking the time to check the technical parts of our paper.. We uploaded a revised version of our paper with additions highlighted in blue. We also respond to a few points below:_
>
> ---
> ### 1. Re: Thm 3.5
> - You're right that $\lVert \theta_a \rVert_2 = \lVert \hat\theta_a \rVert_2 =1$  is a strong assumption. In the current version, we relax it and only assume $\lVert \theta_a \rVert_2 =1$ without loss of generality and no effect to our result.
> - Yes, $\eta_t$ term is multiplied by $B^{-1}$. However, there is no dependence on $B^{-1}$ in the final result. This because the cancellation with the $B$ term from  $\hat\theta =\theta^\top B$.
> - Error Bounds for Thm 3.5: We updated the statement and the proof of Thm 3.5 to incorporate model misspecification. Please see our response to Reviewer BJvW ( Re: 1. Claims of sample efficiency & Re: 2. Thm 3.5 Optimal Pair) for linking offline estimation error to online regret in our setting.
>
> **Changes:**
> - To the statement and proof of Thm 3.5 to add model misspecification.
> - Removed the assumption $\lVert \theta_a \rVert_2 = \lVert \hat\theta_a \rVert_2 =1$
>
> ---
> ### 2. Re: Thm 3.5  Comparison to Latent Bandits Revisited [1]
>
> In [1], the authors prove in Theorem 2 that if deviations in reward estimates are uniformly bounded by a constant $\epsilon$, this adds linear regret to their bound with other terms maintained. We prove an analogous result for CPG. Assuming that for a learned model pair $f, \theta, \forall a, z:\left|\hat{\theta}^{\top} f(g(z))-\theta^{\top} z\right|<\epsilon$, leads to a regret bound for CPG, that is constant when $\epsilon <\Delta_i/2$ and becomes linear when $\epsilon \geq \Delta_i/2$. Resulting in the regret bound:
>
> $$ \mathrm{Reg}_T \leq  \mathcal{1}[\epsilon < \Delta_i/2] \frac{8K\sigma^2\overline{\Delta}_i}{(\Delta_i -2\epsilon)^2} +  \mathcal{1}[\epsilon \geq \Delta_i/2]  \left(\frac{8K\sigma^2\overline{\Delta}_i}{\Delta_i^2} + 2\epsilon T +\frac{\Delta_i T}{2} \right)
> $$
>
>
> In [1] the regret bound depends on $\sqrt{T\log T}$ with a similar $2\epsilon T$ linear term.
> $$\mathrm{Reg}_T\leq T \delta+3|\mathcal{S}|+2 T \varepsilon+2 \sigma \sqrt{6|\mathcal{S}| T \log T}$$
> We summarize the comparison in the following table, and a discussion in the Appendix.
>
>
> |                     | Latent Bandits Revisited                    | ILB                                              |
> | ------------------- | ------------------------------------------- | ------------------------------------------------ |
> | number of arms      | finite                                      | finite                                           |
> | latent state        | discrete < number of arms                   | continuous                                       |
> | context $X_t$       | independent from the latents                | deterministic from latents $X_{i,t} =g(Z_{i,t})$ |
> | reward distribution | sub-Gaussian with $\sigma^2$ proxy variance | Gaussian with $\sigma^2$ variance                |
> | regret              | square-root + linear                                 | constant + linear                                 |
> |                     |                                             |                                                  |
>
> **Changes:**
> - We added a comparison to [1] in the Appendix D, after the proof of Theorem 3.5).
> ---
> ### 3. Re: Identifiability assumptions (infinite per-instance)
>
> Assuming "inifinite per-instance data" is done to reason about _identifiability_, that is: Is the learning strategy (estimand) going to lead us to an optimal model if we are given sufficient data (to estimate it)? There doesn't always exist such a strategy at all, but we show that ours has this property. Proving identifiability is an important step to developing good _estimators_ which can handle different numbers of samples per instance, for example. **Theorem 3.3 makes no claims about statistical estimation, only about identifiability.**  We detail our response further in our response to Reviewer BJvW (Re: Implications of Theorem 3.3).
>
> **Changes:**
> - Changed the presentation of max-likelihood objective (eq. 5)
> ---
> ### 4. Re: About applicable to linear reward & matrix of patient latents
> - You are right that, our theoretical results (Theorem 3.4-5) apply only when the reward is linear with respect to the latent variable $Z$; however, nonlinear w.r.t obserrved contexts, $X_t$. Thus, our framework can handle non-linearities in the data generating process. We also conduct experiments with non-linear reward models in App E.5.
> - Having a $n-$rank matrix of patient latents is a limitation of our work. In practice, $n\leq I$ could be a learned hyper-parameter.
>
> **Additional Changes:**
> - Moved Remark on Theorem 3.4 to main text
> ---
> [1] Hong et al., Latent Bandits revisited, NeurIPS 2020.
>
> ---
> _We hope that our response addresses your concerns and are happy to engage in further discussion! --Authors_

---

### Decision · Action_Editor_yhva · 2026-04-07

**Recommendation:** Accept as is

**Additional Comments:**

After a fruitful discussion with the authors, the reviewers found their concerns to be properly addressed and unanimously recommended accepting the paper. In their views, this is a solid theoretical work with empirical corroboration. While some concern remain over the gap between realistic settings and the strong assumptions made in the paper, I recommend accepting the work. After a brief inspection, I believe the main points in the rebuttal have been already incorporated in the latest version, and I suggest accepting as is.

**Audience:**

Yes

**Audience Explanation:**

The paper can draw the interest of the reinforcement learning and online learning communities.

**Claims And Evidence:**

Yes

**Claims Explanation:**

All the reviewers found the paper to be technically sound, with the theorem statements reported along clear, albeit restrictive, assumptions.